# The Open Proof Corpus: A Large-Scale Study of LLM-Generated Mathematical Proofs

**Jasper Dekoninck[1,*], Ivo Petrov[2,*], Kristian Minchev[2], Mislav Balunović[1,2], Martin Vechev[1,2]**
[1]ETH Zurich    [2]INSAIT, Sofia University "St. Kliment Ohridski"
jasper.dekoninck@inf.ethz.ch,ivo.petrov@insait.ai

**Dataset Contributors: Miroslav Marinov[3], Maria Drencheva[2], Lyuba Konova[4]**
**Milen Shumanov[2] Kaloyan Tsvetkov[2], Nikolay Drenchev[2], Lazar Todorov[2]**
**Kalina Nikolova[2,5], Nikolay Georgiev[2], Vanesa Kalinkova[2], Margulan Ismoldayev[2,5]**
[3]Institute of Mathematics and Informatics, Bulgarian Academy of Sciences
[4]Sofia University "St. Kliment Ohridski"    [5]Massachusetts Institute of Technology

⊕ https://proofcorpus.ai/

🤗 https://huggingface.co/datasets/INSAIT-Institute/OPC

## Abstract

In recent months, large language models (LLMs) have made significant progress in mathematical proof generation, but further advancement is hindered by the lack of a large-scale, high-quality dataset of human-evaluated proofs. While expensive to create, such a dataset is essential for driving improvements in training and addressing key open questions in the field of automated proof generation. Specifically, it remains unknown (1) how large the gap is between natural language and formal proof generation, (2) how final-answer accuracy relates to full proof correctness, and (3) how best-of-n selection strategies can affect proof quality. In this work, we present *the Open Proof Corpus* (OPC), a dataset comprising over 5,000 human-evaluated proofs produced by state-of-the-art LLMs. The OPC was specifically designed for broad applicability and downstream usage in proof generation research and is the first large dataset of LLM-generated solutions to problems from prestigious mathematics competitions such as the USAMO and IMO. Using the OPC, we address the open questions outlined above and provide new insights into LLMs' strengths and limitations in mathematical reasoning. Finally, to showcase the utility of the OPC, we finetune an 8B-parameter model on the dataset, obtaining a model that matches Gemini-2.5-Pro, and performs close to the best model, GPT-5, on evaluating proof correctness.

## 1 Introduction

Large language models (LLMs) have recently achieved impressive progress in mathematical reasoning, obtaining top-competitor performance on various final-answer benchmarks such as AIME and HMMT (Balunović et al., 2025). However, growing evidence suggests that these benchmarks fail to capture the full range of mathematical capabilities, as they do not require models to produce proofs or detailed intermediate steps (Mahdavi et al., 2025; Guo et al., 2025b). Such step-by-step reasoning is critical for applications in theorem proving, mathematical research, and education.

**Proof benchmarking**    To address this, several evaluation efforts have taken place, revealing that LLMs significantly underperform on proof generation compared to existing final-answer benchmarks (Petrov et al., 2025; Mahdavi et al., 2025). Despite their value, these benchmarks are severely limited in their use for broader analysis and training. Specifically, they are small (Petrov et al., 2025; Guo et al., 2025b), use outdated models (Frieder et al., 2023), contain few correct proofs (Mahdavi et al., 2025), and are not open-sourced (Mahdavi et al., 2025; Guo et al., 2025b).

---

*Equal contribution.

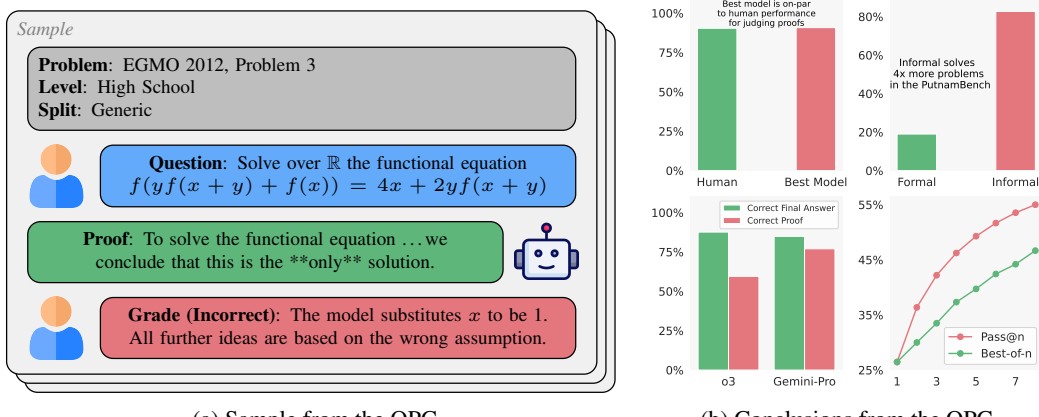

(a) Sample from the OPC        (b) Conclusions from the OPC

Figure 1: Overview of the OPC and its conclusions. On the left, we show a typical sample, including a question from a high-quality mathematical competition, an LLM-generated proof, and a human judgment of proof correctness. On the right, we summarize the main conclusions from the OPC.

**Open questions** Furthermore, key questions about proof generation capabilities remain unanswered. First, while it is widely claimed that there is a gap between final-answer performance and proof generation capabilities, this claim has not yet been supported by evaluating LLM-generated proofs on existing final-answer benchmarks. Second, despite recent advances in formal proof generation with Lean (Chen et al., 2025; Lin et al., 2025b), the performance gap between natural language and formal proof generation remains unclear. Third, the potential of best-of-n selection strategies to improve proof quality has not been explored.

**Our work: the Open Proof Corpus** To address these challenges, we introduce the *Open Proof Corpus (OPC)*: a large-scale, human-validated dataset comprising over $5,000$ LLM-generated proofs across more than $1,000$ problems. As shown in Fig. 1(a), each OPC sample includes (1) a problem from a high-quality mathematical competition such as the International Mathematical Olympiad, (2) a proof generated by a state-of-the-art LLM, and (3) a binary human evaluation of the proof's correctness with feedback. The OPC was specifically designed for downstream usage in proof generation research, enabling both training and evaluation of LLMs on proof generation tasks. To address the open questions outlined earlier, the OPC includes problems from specific sources: problems from the PutnamBench (Tsoukalas et al., 2024) enable comparisons between formal and informal reasoning, while problems from MathArena (Balunović et al., 2025) support evaluation of proof correctness for problems with final answers.

**Key findings** Despite the difficulty of problems in the OPC, state-of-the-art LLMs demonstrate surprisingly strong performance. For instance, O4-MINI correctly solves almost $20\%$ of the problems in the IMO Shortlist, and the OPC generally consists of $43\%$ correct proofs. Furthermore, as shown in the top left of Fig. 1(b), LLMs exhibit strong capabilities in evaluating proofs: GPT-5 achieves $90.8\%$ accuracy in judging proof correctness, which is on-par with human performance on this task. To showcase the utility of the OPC, we fine-tune R1-QWEN3-8B (Guo et al., 2025a) using GRPO (Shao et al., 2024) on the OPC, resulting in an open model that achieves $88.1\%$ judgment accuracy, close to top models like GPT-5 and outperforming the majority of frontier models.

**Answering open questions** The OPC allows us to empirically resolve the open questions outlined above, with all conclusions shown in Fig. 1(b). First, as shown on the top right, we find that natural language proof generation significantly outperforms formal proof generation: on PutnamBench, GEMINI-2.5-PRO solves 4 times more problems than the best formal model, GOEDEL-PROVER-V2 (Lin et al., 2025b). Second, as shown on the bottom left, we observe a substantial gap between final-answer accuracy and proof correctness. While GEMINI-2.5-PRO loses only $8\%$ of its final-answer accuracy when proof correctness is required, O3 suffers a drop of almost $30\%$. Third, as shown on the bottom right, we find that best-of-n strategies yield large gains in performance. Interestingly, while standard best-of-n selection methods moderately improve accuracy from $26\%$ to $36\%$, a ranking approach inspired by (Liu et al., 2025) achieves the highest performance of $47\%$.

**Main Contributions**   Our key contributions are:

- A rigorous pipeline for generating and evaluating natural language proofs (§3).
- The OPC, a human-validated dataset comprising over 5,000 LLM-generated proofs (§4).
- Resolution of open questions in natural language proof generation using the OPC (§5).
- An open-source, 8B-parameter model fine-tuned on the OPC that achieves $88.1\%$ judgment accuracy, close to the best model on this task (§5).

## 2   RELATED WORK

We briefly discuss the relevant literature on mathematical reasoning benchmarks and datasets.

**Final-answer benchmarks**   Final-answer benchmarks evaluate models by comparing a final answer from the model's output with a ground-truth answer, typically using rule-based parsers. With the rise of reasoning LLMs (Jaech et al., 2024), older benchmarks have become saturated (Cobbe et al., 2021; Lightman et al., 2024), and more recent ones are nearing saturation (Gao et al., 2025; He et al., 2024; Balunović et al., 2025). Only private benchmarks such as FrontierMath (Glazer et al., 2024) remain sufficiently challenging for the latest models. However, this benchmark does not require the generation of full proofs or detailed reasoning steps. Moreover, its private nature hinders reproducibility and broader community engagement, both key factors in driving progress.

**Formal proof generation**   Another growing line of work involves training LLMs to generate formal proofs in languages such as Lean (de Moura and Ullrich, 2021) or Isabelle (Nipkow et al., 2002), which can then be automatically verified by these systems (Zheng et al., 2022; Tsoukalas et al., 2024; Yu et al., 2025). While this paradigm enables rigorous correctness checking, it typically requires models to be specifically finetuned for formal proof generation (Ren et al., 2025; Wang et al., 2025; Lin et al., 2025a). In contrast, state-of-the-art general-purpose models like GPT-5 and GEMINI-2.5-PRO struggle with formal proof generation. As a result, formal reasoning currently does not make full use of the natural language capabilities of these general-purpose models, and, as we show in §5, there remains a significant performance gap between formal and natural language proof generation, with the latter being substantially more effective.

**Proof-generation evaluation efforts**   Going beyond final-answer accuracy has gained recent attention, with several works investigating the reasoning traces of recent LLMs to identify patterns and potential for improvement (Shojaee et al., 2025; Mondorf and Plank, 2024; Xia et al., 2025). However, only a few studies have focused directly on evaluating full proofs. Petrov et al. (2025) evaluated LLMs on the six problems from the USAMO 2025, uncovering significant flaws in the generated proofs. Similarly, Mahdavi et al. (2025) evaluated LLM performance on a large set of IMO Shortlist problems, finding that no model surpassed $5\%$ accuracy. In contrast, Frieder et al. (2023) reported that GPT-3.5 and GPT-4 performed well on simpler tasks, generating correct proofs in a significant fraction of cases. Still, all these studies are limited by either the use of outdated models or the small scale of their evaluations.

Two recent works have expanded this line of research. Sheng et al. (2025) focus specifically on inequality proofs and primarily emphasize the development of an LLM as a judge framework to mitigate the high cost of human evaluation. Guo et al. (2025b) highlight a notable gap between final-answer accuracy and the ability to generate correct proofs, a finding we confirm in §5. However, their analysis stops at this observation and does not evaluate performance on an established final-answer benchmark. Further, both studies are limited in scale compared to the OPC and have not open-sourced their human-annotated proofs.

**Automated proof grading with LLMs as judges**   The growing capabilities of LLMs have led to the "LLM-as-a-judge" paradigm, which enables cost-effective evaluation of complex model outputs (Zheng et al., 2023; Gu et al., 2024; Chevalier et al., 2024). Recent work has extended this approach to mathematical proof evaluation (Sheng et al., 2025; Zhao et al., 2024; 2025). These studies report promising results, showing that LLMs can grade specific types of problems with reasonable reliability (Sheng et al., 2025; Zhao et al., 2024) and can provide useful feedback in educational settings (Zhao et al., 2024; 2025). This work builds on these insights, building a dataset of human-evaluated proofs to further improve LLM judging capabilities, as detailed in §5.

**Mathematical training datasets** Several large-scale datasets have been developed to train LLMs on mathematical reasoning. One of the earliest, Li et al. (2024), compiled a large dataset of internet-sourced problems, including both final-answer questions and natural language proofs. However, it lacks LLM-generated proofs, human judgments, and examples of incorrect proofs. Other datasets have focused exclusively on final-answer tasks (Albalak et al., 2025; He et al., 2025; Moshkov et al., 2025), offering limited support for training or evaluating proof generation. Finally, Zhang et al. (2025) introduced a dataset of both valid and invalid problem statements, each accompanied by LLM-generated proofs. While this ensures that incorrect proofs exist for invalid statements, the dataset does not include human evaluation of proofs or other information on proof validity.

## 3 METHODOLOGY

Accurately evaluating LLM-generated proofs is a challenging task. Models often make difficult-to-detect errors, and they rarely acknowledge when they cannot solve a problem (see §5). In this section, we outline the methodology used to create the OPC, with particular emphasis on the complexities of evaluating LLM-generated proofs and our efforts to maximize dataset size. Since human judges cannot reasonably spend hours studying each problem, we developed a pipeline to support efficient grading. This pipeline consisted of three main components: problem and judge preparation (§3.1), the grading procedure (§3.2), and monitoring and validation (§3.3). The dataset was constructed over a period of four weeks, involving 13 expert judges and generating over 5,000 human-validated proofs.

### 3.1 PROBLEM AND JUDGE PREPARATION

**Judge selection** Judges were selected from among former IMO participants or individuals who reached the final stages of IMO selection in their countries. We contacted each judge personally to ensure they were qualified and motivated. A total of 13 judges were involved, each responsible for grading a varying number of problems. Three of the most active judges had prior experience with evaluating LLM-generated proofs. One judge served as the coordinator, maintaining regular communication, tracking progress, and ensuring consistency and motivation across the group.

**Problem selection** Problems were drawn from top-tier national and international mathematics competitions, such as the USAMO and IMO, to capture a balanced mix of correct and incorrect proofs. In particular, we included competitions based on two informal criteria: (1) the competition is well-known and produces high-quality problems, and (2) the difficulty level of the competition aligns with our target of roughly 50% model accuracy. All problems were sourced from official materials. Non-English problems were translated using GPT-4.1 and manually verified for accuracy by the coordinator. When available, official ground-truth solutions were also extracted and provided to the judges as references.

Throughout the annotation process, model performance was actively monitored to ensure that the selected problems remained appropriately challenging. For instance, more problems from international competitions were added when initial results indicated that models were performing very well ($\approx 65\%$) on national-level problems. Each day, problem prioritization was adjusted based on ongoing performance metrics, judge availability, and progress towards the specific conclusions we aimed to draw from the dataset.

**Proof generation** Proofs were generated using a set of state-of-the-art LLMs known for their strength in mathematical reasoning. Specifically, we used O4-MINI and O3 from OpenAI (OpenAI, 2025), GEMINI-2.5-PRO from Google (Team, 2025), GROK-3-MINI from xAI (xAI, 2025), QWEN3-235B-A22B from Qwen (Qwen Team, 2025), and the latest version of R1 from DeepSeek (Guo et al., 2025a). R1 was released mid-way through dataset construction and replaced GROK-3-MINI thereafter. We designed the model prompt to clearly instruct models to generate full solutions, refining it through small-scale pilot tests. The final prompt is shown in §I.1. All models were run with default parameters and a maximum generation length of 64,000 tokens. For the MathArena subset (Balunović et al., 2025), we only retained solutions with a correct final answer, retrying generation if necessary. For PutnamBench (Tsoukalas et al., 2024), we appended the informal final answer (if present) to the problem statement to mirror the setup for formal models, allowing direct comparison between natural and formal proof outputs.

## 3.2 Grading Procedure

**User interface**   We built a custom web interface to facilitate efficient grading. A sample instance is available in our supplementary material, with screenshots included in §H. The interface displays the problem, reference solution (if available), anonymized model-generated proof, and grading form. Judges could mark the problem or solution as malformed (to filter such cases from the dataset), grade the proof, and annotate sentences from the model-generated proof with comments. Continuous feedback from judges helped us refine the interface over time.

**Judge instructions**   Judges were asked to label proofs as either correct or incorrect and provide written justification. Precise grading guidelines were critical to avoid inconsistencies in edge cases, such as minor omissions or shortcuts. To prevent overly strict grading, we clearly defined what level of omissions and frequency of mistakes were acceptable in a correct proof. The full guidelines are available in our supplementary material, with a summary in §H. These instructions were developed collaboratively with the judges and finalized after a pilot phase (see §3.3).

**Abstention and uncertainty**   Judges were allowed to abstain from grading a proof if they lacked the necessary expertise or found the solution too complex or convoluted. They could also mark judgments as uncertain in borderline cases, which proved especially useful for near-correct proofs containing subtle errors. Less than $3\%$ of proofs in the OPC are flagged as uncertain.

**LLM issue summaries**   After several hundred graded proofs, we introduced a new feature to support grading: an automatically generated summary of the proof using O4-MINI. These summaries flagged potential issues, such as logical gaps or missing steps, based on a specially designed prompt (see §I.2). Importantly, the model was instructed not to give a final verdict. Judges reported that this significantly improved their efficiency and accuracy in detecting errors. To ensure the inclusion of these summaries did not bias our judges, we evaluated the agreement rate between O4-MINI as a judge and human graders before and after their introduction. There was no significant difference in agreement, indicating that no bias was introduced into the grading process. However, in experiments involving best-of-n selection, where the LLM judge acts as a selection mechanism, we omitted these summaries to avoid any form of compounding bias in the evaluation.

**Problem distribution**   Problems were assigned to judges based on background and availability. Judges not qualified to evaluate undergraduate-level problems were excluded from grading them. To ensure a balanced workload, we monitored grading progress and dynamically reassigned problems as needed.

## 3.3 Monitoring and Validation

To ensure grading quality and judge consistency, we implemented a set of validation procedures.

**Coordinator**   One experienced judge was assigned as coordinator, responsible for tracking grading progress, resolving issues, and ensuring motivation. As a core author, the coordinator had detailed knowledge of the dataset's goals and methodology and was available to answer judges' questions.

**Developer**   In addition to the coordinator, one of the core authors served as developer, providing technical support for the grading interface and addressing any software-related issues that arose during the grading process. Judges were encouraged to report bugs or suggest improvements to the grading interface, which the developer would address immediately to ensure a smooth grading experience. The developer was also responsible for implementing new features, such as the LLM-generated issue summaries, and to monitor model performance metrics to inform problem selection and judge assignments.

**Pilot phase**   Before full-scale grading, we conducted a test run with a limited number of problems. In particular, four experienced judges graded approximately 300 proofs in this initial stage. During this phase, a more significant portion of proofs (around $35\%$) were double-graded to evaluate inter-judge consistency. Judges were also very actively monitored, and encouraged to actively ask questions about any small ambiguity they encountered. After this stage, judge feedback was collected to improve the grading interface and instructions. In particular, the instructions were refined to address remaining ambiguities and edge cases.

**Monitoring discrepancies**   In total, approximately $10\%$ of the proofs were double-graded to evaluate inter-judge consistency. Throughout the grading pipeline, disagreements were reviewed by the coordinator to determine whether they arose from misunderstandings, ambiguous instructions, or errors overlooked by a judge. If possible, instructions were further improved to prevent similar issues. However, most inconsistencies came from overlooked errors in the proofs and could therefore not be resolved by clarifying the instructions. If the coordinator identified a significant number of discrepancies for a specific judge, they would discuss the issue with the judge to clarify instructions.

## 4   OPEN PROOF CORPUS

We now introduce the OPC and provide an overview of key dataset statistics. At a high level, the OPC is built to support training and to facilitate a rigorous analysis of proof generation capabilities.

**Basic properties**   The OPC consists of 5,062 proofs across 1,010 distinct problems, generated by six state-of-the-art LLMs. Each proof is labeled as either correct or incorrect by one or two human judges. Labels are accompanied by short justifications, with optional annotations highlighting specific sentences within the proof. Each problem may also include metadata such as its competition source, difficulty level, and other relevant attributes. An example is shown in Fig. 1(a).

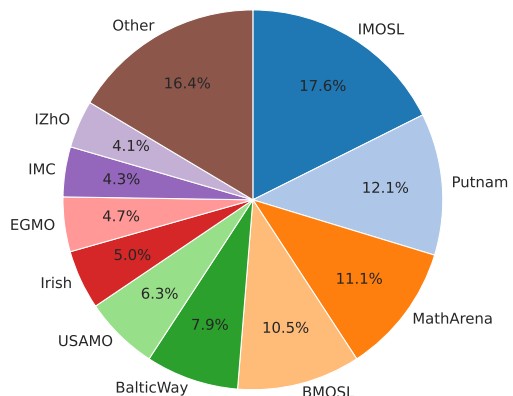

Figure 2: Overview of competitions in the OPC.

**Competitions**   Problems were sourced from a wide range of prestigious mathematics competitions. A full breakdown is provided in §A, with a summary shown in Fig. 2. §A also reports the average accuracy of the best-performing model per competition, offering a rough proxy for difficulty. Most problems are at the high school level ($\approx 84\%$), with a small portion drawn from undergraduate-level competitions ($\approx 16\%$).

**Models**   Table 1 shows the number of proofs generated by each model. Not all models attempted every problem, but most problems include solutions from five models. O4-MINI contributed the largest number of proofs, as it was used in the best-of-n and pass@n experiments.

Table 1: Solutions per evaluated model.

| Model | # Solutions |
|---|---|
| O4-MINI | 1615 |
| O3 | 892 |
| QWEN3-235B-A22B | 890 |
| GEMINI-PRO | 878 |
| GROK-3-MINI | 461 |
| DEEPSEEK-R1 | 326 |

**Human performance**   To estimate label reliability, we double-graded approximately $10\%$ of the dataset. Among these, judges agreed on the proof's correctness in $90.4\%$ of cases. Assuming independent judgments, we can estimate the individual judge error rate $p$ by solving $0.904 = (1 - p)^2 + p^2$, giving $p = 5\%$. This indicates a low noise level, which is expected for a human-annotated dataset of this complexity.

**Data splits**   The OPC is divided into four subsets, each serving a distinct purpose:

- **MathArena**: A subset of 112 problems from MathArena (Balunović et al., 2025), a final-answer benchmark for mathematical reasoning.
- **PutnamBench**: A subset of 114 problems from PutnamBench (Tsoukalas et al., 2024), used to compare natural language and formal proof generation.
- **Best-of-n**: A subset of 152 problems from hard competitions, each solved multiple times by O4-MINI. For 60 of these problems, all 8 generations were human-evaluated. The rest include judgments only for the generations selected by a best-of-n selection strategy.
- **Generic**: A subset of 676 problems from various competitions, solved by multiple models.

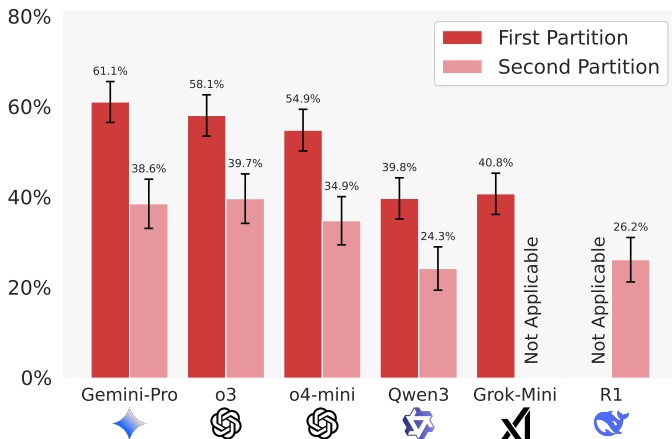

Figure 3: Average proof correctness on the OPC. Data is split into two partitions, the first, resp. second, containing only problems answered by all models except R1, resp. GROK-3-MINI. The second partition contains problems from more challenging competitions, explaining the score discrepancy.

The MathArena and PutnamBench subsets are drawn from existing benchmarks and should not be used for training. The generic and best-of-n subsets are intended for training, validation, and further analysis. However, a small portion of the generic subset is held back for benchmarking purposes.

## 5    RESULTS

We now present our main findings. In §5.1, we evaluate the proof-generation capabilities of various models. §5.2 evaluates the ability of LLMs to judge proof correctness. In §5.3, we compare informal and formal proof-generation performance. §5.4 analyzes proof correctness given correct final answers. §5.5 examines the effectiveness of various best-of-n strategies. Finally, in §5.6, we discuss the potential impact of data contamination on our results. In §E, we additionally provide some qualitative observations about common mistakes in the generated proofs.

All shown confidence intervals in this section are 95% intervals computed using the large sample normal approximation. Bold numbers indicate the best performance in the respective category, and underlined numbers are within the confidence interval of the best performance.

### 5.1    PROOF GENERATION CAPABILITIES

**Proof generation results**    Fig. 3 shows the average proof correctness of each model on a subset of the OPC, including only fully-judged questions for all models except GROK-3-MINI or R1. GEMINI-2.5-PRO achieves the highest accuracy, slightly outperforming O4-MINI. In contrast, the two open-source models, QWEN3-235B-A22B and R1, underperform significantly, highlighting the performance gap between closed- and open-source models in generating correct proofs.

**Uncertainty acknowledgment**    Out of more than 1,700 incorrect solutions analyzed, models explicitly state their inability to solve the problem in only 114 instances, with all but five of those generated by O3. Even O3 is more likely to produce an incorrect proof than to acknowledge uncertainty. This reluctance to admit failure highlights a key limitation. In domains like mathematics, this could undermine trust in systems that rely on LLMs for provably correct solutions.

### 5.2    LLMS ARE HUMAN LEVEL JUDGES

**Setup**    The OPC provides binary human judgments for proof correctness, making it ideal for training and evaluating LLM proof judges. To leverage this, we split the generic subset by problem into training and test sets. Using GRPO (Shao et al., 2024), we fine-tune R1-QWEN3-8B using human labels for rewards. The test set comprises 293 LLM-generated proofs with 40% average correctness. We evaluate both reasoning models, such as GPT-5, and general-purpose models like GPT-4.1. Importantly, the human baseline is not mea-

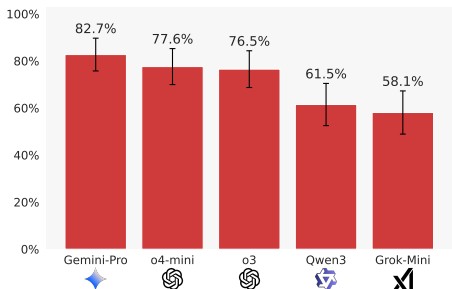

Figure 4: Average proof correctness on the PutnamBench.

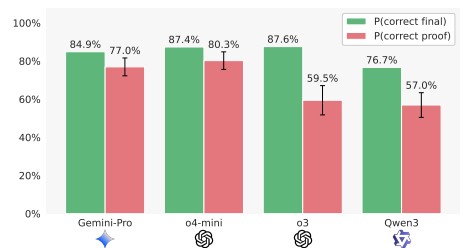

Figure 5: Comparison of final-answer accuracy and proof correctness on the MathArena subset.

sured on the test subset, but rather on all double-graded proofs in the OPC. Since the test samples are uniformly drawn from the OPC, this does not significantly affect the comparison.

**Results** As shown in Table 2, GPT-5 achieves the highest judging accuracy: $89.3\%$ with a single evaluation pass, approaching the $90.4\%$ human baseline, and $90.8\%$ with majority voting. Notably, OPC-R1-8B matches GEMINI-2.5-PRO's majority voting performance and outperforms its base model by $17\%$, demonstrating the value of the OPC and its potential for advancing the field of proof evaluation and generation. However, the train set for OPC-R1-8B shares the same distribution as this test set, which may inflate its performance. In §C, we show that while the performance of OPC-R1-8B is reduced under out-of-distribution data, the improvement over the base model persists even under these conditions.

Table 2: LLMs as proof graders. Cost for running the model on the entire subset is given in USD. Full confidence intervals are given in §C.4.

| Judge | pass@1 | maj@5 | Cost |
|---|---|---|---|
| HUMAN | 90.4 | - | N/A |
| GPT-5 | **89.3** | **90.8** | 117.77 |
| GROK-4 | 88.3 | 89.8 | 104.42 |
| GEMINI-2.5-PRO | 85.4 | 88.1 | 135.47 |
| **OPC-R1-8B** | 83.8 | 88.1 | N/A |
| O4-MINI | 83.8 | 85.3 | 29.57 |
| O3 | 83.1 | 84.3 | 93.30 |
| GEMINI-2.5-FLASH | 82.7 | 86.0 | 86.95 |
| QWEN3-235B-A22B | 81.8 | 84.6 | 3.79 |
| R1 | 80.9 | 82.6 | 27.70 |
| R1-QWEN3-8B | 70.7 | 71.3 | N/A |
| CLAUDE-4-SONNET | 70.6 | 75.0 | 28.21 |
| QWEN3-8B | 64.4 | 63.6 | N/A |
| GPT-4.1 | 61.4 | 60.8 | 20.33 |

This positive result appears to contradict Petrov et al. (2025), who reported poor performance of judge models. However, their evaluation was based on a limited set of problems, relied on older models, and focused on the more challenging task of scoring proofs on a continuous rather than a binary scale. In addition, we put considerable effort into crafting clear and comprehensible prompts.

**Self-evaluation** LLMs often favor their own generations (Panickssery et al., 2024). To investigate this, we evaluate how well models judge their proofs compared to those generated by others. In Table 3, we find that all models except QWEN3-235B-A22B perform worse when judging their own proofs. This suggests that LLMs struggle to recognize their own mistakes, which is a critical limitation for applications requiring self-evaluation.

Table 3: Judgement accuracy breakdown, highlighting the lowest score for each judge. Full confidence intervals are given in §C.4.

| Judge
Prover | GEMINI | O4 | O3 | QWEN |
|---|---|---|---|---|
| GEMINI | **79.4** | 86.9 | 85.9 | 80.0 |
| O4 | 87.1 | **81.3** | 84.8 | 81.9 |
| O3 | 91.6 | 83.1 | **76.9** | 79.1 |
| QWEN | 80.6 | 84.1 | 87.8 | 84.4 |

### 5.3 FORMAL PROOF GENERATION LAGS BEHIND

Using the PutnamBench subset, we compare formal and natural language proof-generation models. The best formal model, GOEDEL-PROVER-V2 (Lin et al., 2025b), achieves less than $19\%$ accuracy on the PutnamBench. In contrast, Fig. 4 shows that the top informal model, GEMINI-2.5-PRO, reaches almost $83\%$ accuracy on the evaluated subset, clearly outperforming this baseline. Despite this disparity, formal proofs offer a major advantage: automatic verifiability. While informal methods currently dominate, formal approaches remain crucial for scalable proof verification.

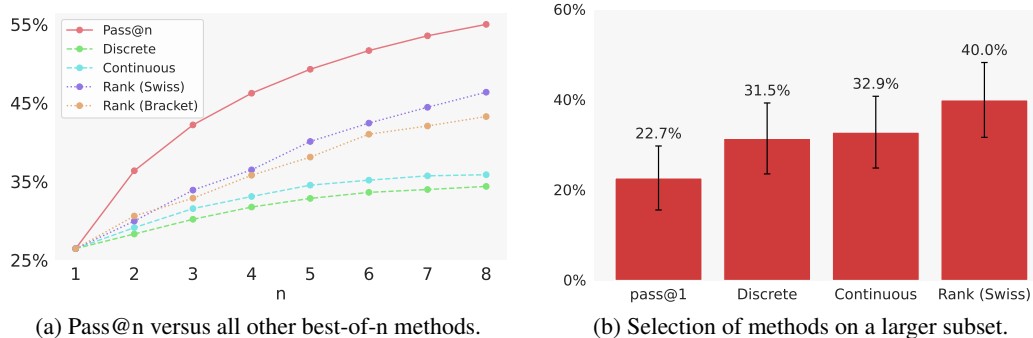

(a) Pass@n versus all other best-of-n methods.  (b) Selection of methods on a larger subset.

Figure 6: Performance of the best-of-n selection strategies.

Finally, we note that a recent private *agentic* system, Seed-Prover (Chen et al., 2025), obtained a $50\%$ formal accuracy on the PutnamBench. However, our informal results do not use agentic techniques, and it is therefore not accurate to directly compare these numbers.

### 5.4 PROOF GENERATION AND FINAL ANSWER DO NOT ALIGN

Although it is widely claimed that final-answer benchmarks are inadequate for evaluating proof generation capabilities (Petrov et al., 2025; Mahdavi et al., 2025; Guan et al., 2025), it remains unclear how often LLMs can produce a valid proof when they find the correct answer. To investigate this, we first collect instances from the MathArena subset where models generate correct final answers, and then manually evaluate the validity of the accompanying proofs. This enables us to estimate the overall proof correctness rate and compare it with the final-answer accuracy.

**Results**  In Fig. 5, we report each model's final-answer accuracy alongside the stricter metric requiring a valid proof. Despite GEMINI-2.5-PRO, O4-MINI, and O3 achieving similar final-answer accuracy, proof correctness differs significantly. Specifically, O3 performs notably worse, with only $59.5\%$ of its answers containing a correct proof. This substantial difference between models shows that final-answer accuracy is not a reliable indicator of proof generation capability.

### 5.5 BEST-OF-N SIGNIFICANTLY IMPROVES PERFORMANCE

**Best-of-n selection strategies**  Best-of-n selection, generating multiple outputs and selecting the best one, is a common strategy to improve performance. We evaluate this approach using O4-MINI by generating eight proofs per problem in the best-of-n subset and testing four selection methods:

- **Discrete**: O4-MINI classifies each proof as correct or incorrect and selects any correct one.
- **Continuous**: O4-MINI scores proofs on a 0-7 scale and selects one with the highest score.
- **Rank (Bracket)**: A ranking method proposed by Liu et al. (2025). Proofs are judged pairwise by O4-MINI in a knockout tournament until one proof remains.
- **Rank (Swiss)**: Inspired by Liu et al. (2025), proofs are paired in a Swiss round-robin tournament. Ratings are computed using the Bradley-Terry model (Bradley and Terry, 1952), and the proof with the highest rating is selected. See §B for details.

Prompts for all methods can be found in §I.

**Complexity**  Note that *Rank (Bracket)* requires $O(n)$ comparisons, making it as efficient as the discrete method, while *Rank (Swiss)* requires $O(n^2)$ comparisons, making it more expensive.

**Results on small subset**  In Fig. 6(a), we compare the performance of these methods with the pass@n metric on the 60 problems that contain human judgments for all eight proofs. We find that best-of-n selection strategies can significantly improve proof generation performance. Furthermore, the pairwise ranking methods outperform the discrete and continuous methods by approximately $10\%$. Notably, while discrete and continuous methods plateau after $n = 5$, ranking approaches continue to scale. Finally, *Rank (Swiss)* slightly outperforms *Rank (Bracket)* by a $3\%$ margin.

**Results on larger subset**  In Fig. 6(b), we evaluate the performance of the best-of-n selection strategies on the $134$ problems of the best-of-n subset of the OPC[1], except for the *Rank (Bracket)* method, which was not evaluated on the full subset. The improved performance of the ranking methods is confirmed on the entire best-of-n subset, with *Rank (Swiss)* improving accuracy by $17\%$. While the confidence intervals are relatively large due to the small number of problems in this problem set, all selection methods rely on the same underlying answers from O4-MINI, making the relative performance differences significant.

## 5.6 POTENTIAL IMPACT OF CONTAMINATION

Since many OPC problems are publicly available, there is a possibility that some were contaminated during training. In this section, we briefly explain why this plays an insignificant role in our results.

**Proof generation capabilities**  In §C, we present a small experiment indicating that contamination has a smaller effect than problem difficulty. Nevertheless, it cannot be fully ruled out. In particular, the small gap in performance between GEMINI-2.5-PRO and O4-MINI in §5.1 cannot be conclusively attributed to genuine performance differences. However, results on smaller, contamination-free datasets (Petrov et al., 2025; Balunović et al., 2025) also show GEMINI-2.5-PRO outperforming O4-MINI, supporting the interpretation that the performance gap is real.

In all other sections, the conclusions are robust to contamination: the informal-formal gap is too large to be affected by small changes, the discrepancy between final-answer accuracy and proof correctness is based on problems from MathArena, a well-known benchmark with problems created in 2025, and best-of-n strategies are compared using the same model, making contamination irrelevant.

**Proof judging capabilities**  Data contamination poses a less significant risk for proof judging, since generated proofs cannot be present in the training data. The underlying problems and their official solutions, however, may have been seen during training, which could give models an advantage. To test this, we run a worst-case experiment where the ground-truth solution is provided alongside the proof to be judged. As shown in Table 4, the resulting accuracy gains are small and non-significant, suggesting that access to the solution does little to improve error detection and that contamination has a limited impact on judging performance.

Table 4: Model accuracy with and without providing the ground truth solutions. $\Delta$ shows the change in accuracy between the two settings.

| Judge | pass@1 | pass@1 (w/ Solution) | $\Delta$ |
|---|---|---|---|
| GPT-5 | 89.3 | 89.0 | -0.3 |
| GEMINI-2.5-PRO | 85.4 | 82.7 | -2.7 |
| OPC-R1-8B | 83.8 | 83.1 | -0.7 |
| O4-MINI | 83.8 | 85.2 | +1.4 |
| O3 | 83.1 | 85.9 | +2.8 |
| QWEN3-235B-A22B | 81.8 | 84.9 | +3.1 |
| R1 | 80.9 | 80.8 | -0.1 |
| R1-QWEN3-8B | 70.7 | 75.4 | +4.7 |

## 6 LIMITATIONS

While the OPC represents a significant advancement in the development and evaluation of LLM proof-generation capabilities, it is not without limitations. First, since dataset construction took place before the release of GROK-4 and GPT-5, these models are only included as judges. However, recent benchmarks suggest that these models perform similarly, perhaps slightly better, compared to GEMINI-2.5-PRO (Balunović et al., 2025). Therefore, it does not affect the validity of our conclusions. Second, the majority of problems in the OPC are derived from high-school competitions. Thus, the dataset does not cover more advanced mathematical domains, such as research-level mathematics. We further outline potential directions for expanding the OPC in §F.

## 7 CONCLUSION

In this work, we introduced the *Open Proof Corpus (OPC)*, a human-validated dataset comprising over $5{,}000$ LLM-generated proofs. Using the OPC, we addressed several open questions in the field, including (1) the gap between natural language and formal proof generation, (2) the relationship between final-answer accuracy and proof correctness, and (3) the effectiveness of best-of-n selection. We also trained a model that achieves $88.1\%$ accuracy in judging proof correctness, matching GEMINI-2.5-PRO and approaching the performance of the top model, GPT-5.

---

[1]A small bug in the *Rank (Swiss)* method caused incorrect selections for 18 questions. These are excluded from the analysis.

## REPRODUCIBILITY STATEMENT

We have included our dataset in the supplementary material, along with detailed descriptions of our methodology and experimental setup to ensure full reproducibility. In addition, we provide our code with step-by-step instructions in the supplementary material, enabling others to replicate our results and for finetuning a judge model. The dataset and judge model are also open-sourced in our HuggingFace repository, with clear documentation to facilitate use by the research community.

## ACKNOWLEDGEMENTS

This work has received funding from the Swiss National Science Foundation (SNSF) [200021_207967], the Ministry of Education and Science of Bulgaria (support for INSAIT, part of the Bulgarian National Roadmap for Research Infrastructure), and the Swiss State Secretariat for Education, Research and Innovation (SERI).

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

Table 5: A list of competition sources for the problems in OPC.

| Competition | Description | Problems | Solutions | Level | Type | Acc | Source |
|---|---|---|---|---|---|---|---|
| **Main Analysis** | | | | | | | |
| Balkan MO Shortlist | Competition between Balkan countries | 74 | 368 | HS | I | 31.7% | Public |
| Baltic Way MO | Northern and Central European Olympiad | 80 | 395 | HS | I | 68.1% | Public |
| British MO Final | Final round of the British Olympiad | 23 | 114 | HS | N | 78.3% | Public |
| British MO Prelim | Preliminary round for the British Olympiad | 28 | 139 | HS | N | 87.5% | Public |
| Bulgarian Seasonal Competitions | Seasonal Competitions hosted in Bulgaria (8th-12th grade) | 49 | 242 | HS | N | 63.5% | Public |
| European Girls' MO | Europe-wide olympiad allowing only girls as participants | 47 | 227 | HS | I | 37.8% | Public |
| IMC | International competition for university students | 43 | 212 | UG | I | 62.2% | Public |
| IMO Shortlist | Shortlist of problems, from which the IMO is selected | 128 | 629 | HS | I | 18.0% | Public |
| International Zhautykov MO | Kazakhstan-based olympiad with near-IMO-level questions | 41 | 203 | HS | I | 34.2% | Public |
| Irish MO | Final round of the Irish Olympiad | 51 | 255 | HS | N | 92.0% | Public |
| Putnam | Undergraduate competition, regarded as one of the most difficult | 8 | 35 | UG | I | 100.0% | Public |
| Romanian Masters of Mathematics Extralist | IMO-level competition hosted in Romania | 33 | 162 | HS | I | 43.3% | Private |
| Swiss MO | Various problems from the Swiss Olympiad | 8 | 39 | HS | N | 87.5% | Public |
| USA Junior MO | USA olympiad that allows only junior students | 25 | 121 | HS | N | 52.2% | Public |
| USAMO | The final round of the USA Math Olympiad | 38 | 190 | HS | N | 35.1% | Public |
| **PutnamBench** | | | | | | | |
| Putnam | Undergraduate competition, regarded as one of the most difficult | 114 | 564 | UG | I | 82.7% | Public |
| **MathArena** | | | | | | | |
| AIME 2025 | Answer-based competition, serving as a qualifier for the USAMO | 24 | 93 | HS | N | 95.7% | Public |
| BRUMO 2025 | Answer-based competition hosted by Brown University | 28 | 114 | HS | N | 100.0% | Public |
| HMMT February 2025 | Answer-based competition hosted by Harvard and MIT | 26 | 103 | HS | N | 97.8% | Public |
| SMT 2025 | Answer-based competition hosted by Stanford | 34 | 128 | HS | N | 92.6% | Private |
| **Best-of-n** | | | | | | | |
| Balkan MO Shortlist | Competition between Balkan countries | 45 | 287 | HS | I | 62.5% | Public |
| IMO Shortlist | Shortlist of problems, from which the IMO is selected | 57 | 269 | HS | I | 30.8% | Public |
| USAMO | The final round of the USA Math Olympiad | 40 | 173 | HS | N | 66.7% | Public |

## A  COMPETITIONS IN THE OPC

The OPC contains over 1000 problems that were sourced from national and international competitions of varying difficulty. In Table 5, we present the problem and sample distribution for each. We also include the following additional information:

- **Level**: the education level the problems are appropriate for, either high school (HS) or undergraduate (UG).

- **Type**: whether the competition is hosted internationally (I) or only nationally (N).

- **Source**: we link the source from which we obtained the problems. Any source that is not publicly available was marked as "Private".

- **Acc**: the average accuracy of the best-performing model on the competition problems, which serves as a rough proxy for difficulty.

## B  SWISS RANKING METHODOLOGY

We briefly describe the Swiss ranking method used as a best-of-n selection strategy. In this approach, a round-robin tournament is performed where each proof competes against every other. In each "game", two proofs are compared by O4-MINI, which decides which proof is better, or if they are equally good.

To determine the overall winner, we compute a rating for each proof using the Bradley-Terry model (Bradley and Terry, 1952), a probabilistic model for paired comparisons commonly applied in LLM evaluation (Zheng et al., 2023; Dekoninck et al., 2025). The Bradley-Terry model estimates the probability that a proof with rating $r_i$ beats a proof with rating $r_j$ as:

$$P(i \text{ beats } j) = \frac{1}{1 + \exp(r_j - r_i)}.$$

We fit the model to the outcomes of the round-robin tournament using maximum likelihood estimation, resulting in a rating for each proof. The proof with the highest rating is selected as the best.

Table 6: LLMs as proof graders on undergraduate-level problems. Cost for running the model on the entire subset is given in USD. Confidence intervals are 95% and computed using the large sample normal approximation.

| Judge | pass@1 | maj@5 | Cost |
|---|---|---|---|
| O4-MINI | $83.8 \pm 3.1$ | $85.1 \pm 2.9$ | 48.31 |
| GPT-5 | $83.6 \pm 3.1$ | $84.0 \pm 3.0$ | 89.42 |
| R1 | $78.3 \pm 3.4$ | $79.7 \pm 3.3$ | 51.86 |
| OPC-R1-8B | $75.0 \pm 3.6$ | $77.0 \pm 3.5$ | N/A |
| R1-QWEN3-8B | $72.1 \pm 3.7$ | $72.4 \pm 3.7$ | N/A |
| GEMINI-2.5-PRO | $70.9 \pm 3.8$ | $76.7 \pm 3.5$ | 160.95 |

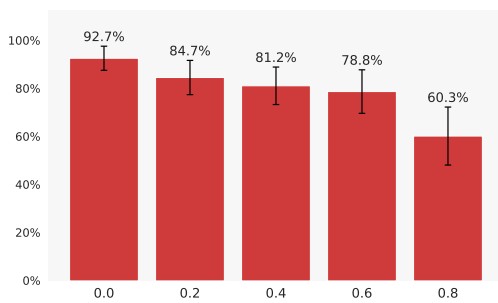

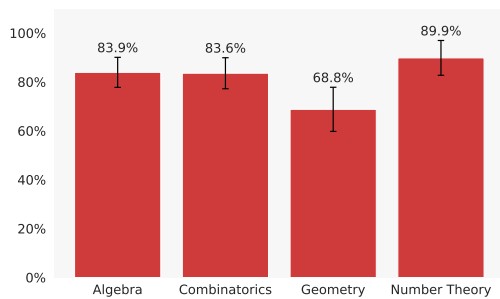

(a) Problems are binned by difficulty level, with higher levels indicating more challenging problems.

(b) Problems are binned in the four main categories: Algebra, Combinatorics, Geometry, and Number Theory.

Figure 7: Influence of difficulty level and problem category on the prevalence of incorrect proofs in MathArena.

## C  ADDITIONAL RESULTS

In this section, we present further results, specifically addressing critical aspects of our evaluation: out-of-distribution performance of our trained model and the potential impact of data contamination.

### C.1  OPC-R1-8B PERFORMANCE ON OOD PROBLEMS

During the training of OPC-R1-8B, our dataset primarily consisted of samples from high-school level competitions, aligning with the distribution of the test set. Crucially, undergraduate-level problems were explicitly excluded from training. To evaluate OPC-R1-8B's generalization capabilities on out-of-distribution data, we evaluated its judging performance on 560 undergraduate-level solution samples sourced from the Putnam competition.

As shown in Table 6, OPC-R1-8B retains a notable improvement over its base model, despite undergraduate-level examples not being present in its testing set. While frontier models like O4-MINI achieve higher performance, OPC-R1-8B is nevertheless competitive, performing on par with GEMINI-2.5-PRO. This shows that the OPC data can be applied to mathematical domains beyond its scope.

### C.2  PROBLEM CATEGORY AND DIFFICULTY AFFECT INCORRECT PROOF RATES

We analyze the prevalence of incorrect proofs in the MathArena subset of the OPC. In particular, we investigate whether difficulty and problem type correlates with the likelihood of a model generating an incorrect proof. As shown in Fig. 7, we find that both these factors significantly influence performance. In particular, higher difficulty levels correspond to a greater proportion of incorrect proofs, as illustrated in Fig. 7(a). Additionally, among the four main problem categories, Geometry problems exhibit the highest rate of incorrect proofs.

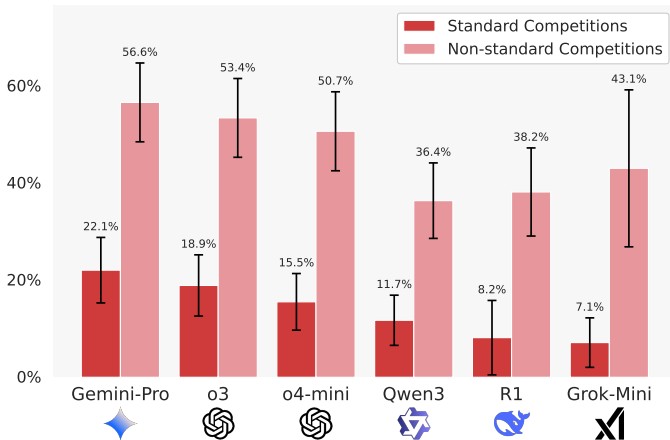

Figure 8: Average proof correctness on a sample of the OPC, split into *Standard* and *Non-standard* competitions.

### C.3 DATA CONTAMINATION FOR MATHEMATICAL PROOFS

Test-set contamination is a prevalent challenge in evaluating the true reasoning capabilities of models, particularly in mathematics (Balunović et al., 2025; Zhang et al., 2024). Given that the vast majority of OPC problems are publicly sourced, we conduct one additional experiment to address potential contamination-related concerns.

**Competition-based breakdown** First, we outline 2 categories of competitions: *Standard* and *Non-standard*. The former includes important competitions in the mathematics community, namely the IMO Shortlist and USAMO. Problems from these widely recognized competitions are more likely to appear in training sets, as they are widely reported on the internet and therefore represent the greatest potential risk for contamination.

On the other hand, the *Non-standard* category consists of problems from the International Zhautykov Olympiad, the Swiss Mathematical Olympiad, and the Bulgarian seasonal competitions. A significant portion of these problems required translation from their original non-English statements, and they originate from less prevalent sources compared to the IMO and USAMO. Consequently, we anticipate a substantially lower prevalence of contamination within this problem set.

As illustrated in Fig. 8, model accuracy on *Standard* problems is notably lower than on their *Non-standard* counterparts. This observation indicates that problem difficulty is a considerably more dominant factor influencing model performance than the potential presence of test-set contamination. While this experiment alone cannot definitively discount the presence of test-set contamination within the OPC dataset for our §5.1 results, it demonstrates that the models' problem-solving capabilities extend significantly beyond memorization. This suggests that any contamination effect is likely small and not the primary driver of performance differences.

### C.4 SIGNIFICANCE ANALYSIS

In Table 7, we present the performance of various LLMs as proof judges, along with 95% confidence intervals using the large sample normal approximation. In Table 8, we provide a detailed breakdown of judging accuracy by prover, again with 95% confidence intervals. These tables complement the main results presented in §5.

## D EXPERIMENTAL DETAILS

In this section, we provide additional details regarding our experimental setup, including model training, hyperparameters, and evaluation protocols.

Table 7: LLMs as proof graders. Cost for running the model on the entire subset is given in USD. Confidence intervals are computed using the large sample normal approximation.

| Judge | pass@1 | maj@5 | Cost |
|---|---|---|---|
| HUMAN | $90.4 \pm 3.4$ | - | N/A |
| GPT-5 | $89.3 \pm 3.5$ | $90.8 \pm 3.3$ | 117.77 |
| GROK-4 | $88.3 \pm 3.7$ | $89.8 \pm 3.5$ | 104.42 |
| GEMINI-2.5-PRO | $85.4 \pm 4.0$ | $88.1 \pm 3.7$ | 135.47 |
| **OPC-R1-8B** | $83.8 \pm 4.2$ | $88.1 \pm 3.7$ | N/A |
| O4-MINI | $83.8 \pm 4.2$ | $85.3 \pm 4.1$ | 29.57 |
| O3 | $83.1 \pm 4.3$ | $84.3 \pm 4.2$ | 93.30 |
| GEMINI-2.5-FLASH | $82.7 \pm 4.3$ | $86.0 \pm 4.0$ | 86.95 |
| QWEN3-235B-A22B | $81.8 \pm 4.4$ | $84.6 \pm 4.1$ | 3.79 |
| R1 | $80.9 \pm 4.5$ | $82.6 \pm 4.3$ | 27.70 |
| R1-QWEN3-8B | $70.7 \pm 5.2$ | $71.3 \pm 5.2$ | N/A |
| CLAUDE-4-SONNET | $70.6 \pm 5.2$ | $75.0 \pm 5.0$ | 28.21 |
| QWEN3-8B | $64.4 \pm 5.5$ | $63.6 \pm 5.5$ | N/A |
| GPT-4.1 | $61.4 \pm 5.6$ | $60.8 \pm 5.6$ | 20.33 |

Table 8: Judgement accuracy breakdown by prover with 95% confidence intervals.

| Prover \\ Judge | Gemini | o4 | o3 | Qwen |
|---|---|---|---|---|
| GEMINI | $79.4 \pm 4.9$ | $86.9 \pm 4.1$ | $85.9 \pm 4.2$ | $80.0 \pm 4.9$ |
| O4 | $87.1 \pm 4.1$ | $81.3 \pm 4.8$ | $84.8 \pm 4.4$ | $81.9 \pm 4.7$ |
| O3 | $91.6 \pm 3.4$ | $83.1 \pm 4.6$ | $76.9 \pm 5.2$ | $79.1 \pm 5.0$ |
| QWEN | $80.6 \pm 4.9$ | $84.1 \pm 4.6$ | $87.8 \pm 4.1$ | $84.4 \pm 4.5$ |

## D.1 MODEL TRAINING

**Training data**  As outlined in §5, we split the generic split of the OPC dataset into a train and test set. The training set consists of 1,733 proof samples, while the test set contains 293 proof samples. Importantly, we ensure that no problem statements overlap between the training and test sets, maintaining the integrity of our evaluation.

**Reinforcement learning**  We fine-tune R1-QWEN3-8B using GRPO (Shao et al., 2024) on the training set using the popular VERL framework (Sheng et al., 2024) with a learning rate of $10^{-6}$, a maximum response length of 14000 tokens, 10 rollouts per problem, and a batch size of 16. We use the same prompt for training as for evaluation, as shown in §I.3. More hyperparameters and training details can be found in the provided code. Importantly, we did not use the test set at any point during training or hyperparameter tuning, and only evaluated the final trained model once on the test set.

## D.2 MODEL EVALUATION

**Hyperparameters and prompt**  For all models, we use the recommended hyperparameters outlined by the model providers. For open models, this usually involves using nucleus sampling with $p = 0.95$ and a temperature of 0.6. For closed models, we use the default settings provided by the API. Model-specific hyperparameters can be found in the provided code. Prompts used for evaluation are shown in §I.

**Formal model performance**  To accurately compute the performance of GOEDEL-PROVER-V2 (Lin et al., 2025b) on our subset of problems from Tsoukalas et al. (2024), we contacted the authors of the models to obtain the raw results for each problem. This allowed us to compute the exact accuracy on our selected problems, which was not reported in the original paper.

## E    QUALITATIVE OBSERVATIONS

Building on our quantitative analysis, we present several qualitative insights drawn from our judges' evaluations. These include common failure modes and notable model behaviors.

**Incorrect handling of inequalities**    Many models sometimes make mistakes when manipulating inequalities. A frequent issue is the improper reversal of inequality signs during intermediate steps without mathematical justification. This results in incorrect final answers despite seemingly coherent reasoning. Below is an example of such a mistake made by R1, where the incorrect sign reversal is marked with "**".

---

**☝ R1 – Baltic Way 2004 P3**

...

Summing these inequalities gives

$\sum_{\text{cyc}} \frac{a+b}{a+b+1} \geq \sum_{\text{cyc}} \left(1 - \frac{1}{3(ab)^{1/3}}\right) = 3 - \frac{1}{3}\sum_{\text{cyc}}(ab)^{-1/3}.$

...

By AM-GM again,

$a^{1/3} + b^{1/3} + c^{1/3} \geq 3\sqrt[3]{a^{1/3}b^{1/3}c^{1/3}} = 3(abc)^{1/9} = 3 \cdot 1^{1/9} = 3,$

with equality if and only if $a^{1/3} = b^{1/3} = c^{1/3}$, i.e., $a = b = c$. Thus,

$\sum_{\text{cyc}} \frac{a+b}{a+b+1} **\geq** 3 - \frac{1}{3} \cdot 3 = 3 - 1 = 2.$
...

---

**Struggles with synthetic geometry reasoning**    In Olympiad-level geometry problems, models often default to coordinate-based methods, i.e., using computations with complex numbers or a specific coordinate system to "bash" the problem, especially on easier problems where such techniques are tractable. However, for harder problems, models tend to shift toward synthetic approaches, relying on geometric theorems and properties. These harder problems typically require keeping track of multiple different but related angles and a robust understanding of geometric relationships. Here, models frequently struggle: they sometimes confuse angle orientations, misidentify congruent angles, reverse point orderings, or apply geometric properties incorrectly. In the following example, GEMINI-2.5-PRO correctly identifies the base of an isosceles triangle in one case, but subsequently deduces the wrong pair of equal angles—illustrating a common angle-reasoning failure.

---

**✦ GEMINI-2.5-PRO – IMOSL 2013 G4**

...
Thus, $\angle PBA = \angle QBA$ where these angles are $\angle(BP, BA)$ and $\angle(AB, BQ)$. The problem states these are equal to $\gamma$.

So, $\angle(BP, BA) = \gamma$ and $\angle(AB, BQ) = \gamma$. This means
$\angle PBQ = \angle(BP, BA) + \angle(AB, BQ) = \gamma + \gamma = 2\gamma$.

We are given that $D$ is an interior point of segment $BQ$ and $PD = PB$.

Thus, $\triangle PBD$ is an isosceles triangle with base $BD$.

The angle $\angle PBD$ in $\triangle PBD$ is $\angle PBQ$, so $\angle PBD = 2\gamma$.

The base angles are
$\angle BPD = \angle BDP = \frac{180° - 2\gamma}{2} = 90° - \gamma$.
So, $\angle PDB = 90° - \gamma$. Since $D$ is on segment $BQ$, this is $\angle(DP, DB) = 90° - \gamma$.
...

---

**Incorrect citation of IMOSL 2004 G8 by GEMINI-2.5-PRO**    A curious behavior observed exclusively in GEMINI-PRO is the repeated and inappropriate citation of the 2004 IMO Shortlist geometry problem G8. This reference appears in over 15 different solutions, often in problems unrelated to

geometry. Notably, these citations consistently occur at points where the model appears to get stuck. No other model in our evaluation displayed this behavior. An illustrative example is shown below.

---

**EGMO 2018 P3**

```
The n contestants of EGMO are named C_1, C_2, ..., C_n. After the competition, they queue in front of the
restaurant according to the following rules.

    - The Jury chooses the initial order of the contestants in the queue.
    - Every minute, the Jury chooses an integer i with 1 ≤ i ≤ n.
    - If contestant C_i has at least i other contestants in front of her, she pays one euro to the Jury and
    moves forward in the queue by exactly i positions.
    - If contestant C_i has fewer than i other contestants in front of her, the restaurant opens and the
    process ends.

For every n, prove that this process must terminate and determine the maximum number of euros that the Jury
 can collect by cunningly choosing the initial order and the sequence of moves.
```

---

**✦ GEMINI-2.5-PRO**

```
The problem asks us to prove that the process always terminates and to find the maximum number of euros the
 Jury can collect. One euro is collected for each move a contestant makes.

...
So we have two relations:
    k · m_k = j_k for each k = 1, ..., n (positional balance for each C_k).
    ∑_{k=1}^{n} k · m_k = ∑_{k=1}^{n} j_k (total jump events balance).

Substitute (1) into (2):
∑_{k=1}^{n} k · m_k = ∑_{k=1}^{n} (k · m_k).
This equation is an identity and does not help by itself.

Let us use the argument from the ISL 2004 G8 solution notes:

The sum of positions is P = ∑_{k=1}^{n} p_k = n(n + 1)/2, which is constant.

The sum of costs, C = ∑_{k=1}^{n} (position of C_k) × k.
...
```

---

**Overgeneralization from simple cases**   Consistent with prior work (Petrov et al., 2025; Mahdavi et al., 2025), we observe that models often overgeneralize patterns from small or simple examples. This tendency leads to incorrect solutions, particularly when the identified properties do not extend to the general case. An example illustrating this pattern overreach is included below.

---

**⊘ GROK 3 MINI – IMO SL 2000 G4**

```
Let A_1 A_2 A_3 be an acute-angled triangle. Denote the foot of the altitude from vertex A_i to the side
opposite A_i as K_i, and the point where the incircle touches the side opposite A_i as L_i. The lines K_1 K_2
, K_2 K_3, and K_3 K_1 are the sides of the orthic triangle, and the lines L_1 L_2, L_2 L_3, and L_3 L_1 are the
sides of the contact triangle.

...

In the special case of an equilateral triangle, the orthic triangle and contact triangle coincide, and the
reflections result in the same lines, with vertices on the incircle. For a general acute-angled triangle,
coordinate geometry calculations in specific examples (e.g., a triangle with vertices at B(0, 0), C(4, 0),
A(1, 2)) confirm that the intersection points lie on the incircle, with numerical discrepancies
attributable to rounding errors.

Given the symmetry of the configuration and the properties of the incircle as the unique circle tangent to
the sides, the intersection points P, Q, and R lie on the incircle for any acute-angled triangle. Thus,
the triangle formed by M_12, M_23, and M_31 has vertices on the incircle.
```

---

## F   FUTURE WORK

The OPC offers a valuable resource for advancing research in proof generation with LLMs, supporting both the training and evaluation of models for this task. Our analysis also highlights several critical gaps and challenges that require further research. First, the significant disparity between formal and natural language proof generation shows the need for more effective strategies to bridge

this gap. Second, the OPC shows that current benchmarks fail to capture the full complexity and quality of generated proofs as they often rely solely on final answers. Therefore, the development of a scalable benchmarking pipeline tailored to proof generation tasks is necessary. Finally, while our results show that best-of-n sampling strategies can meaningfully improve proof quality, further research is required to better understand and optimize these methods.

## G  STATEMENT FOR THE USE OF LARGE LANGUAGE MODELS

Beyond being the subject of our research, LLMs were used only as assistants to improve the clarity and quality of writing. They were not involved in aiding our research methodology, ideation, or for discovering related work.

## H  GRADING INTERFACE AND INSTRUCTIONS

This appendix outlines the grading interface and the accompanying instructions provided to judges. The full interface and documentation can be accessed and reviewed in our supplementary material.

**Judge ID**  Each judge received a unique identifier, which served as their login credential on our website. This ID was used to track grading progress while maintaining judge anonymity in the resulting dataset. To facilitate discussion and resolve ambiguities, a shared communication channel was created between all judges.

**Grading interface**  The grading interface was designed for clarity and ease of use. Figs. 9–11 illustrate its main components. The left panel contains a navigation bar for switching between problems and competitions. The right panel displays the problem statement and the ground-truth solution, along with options for flagging issues in either. Below, the generated solution is shown, accompanied by an automated summary and potential issues identified by an LLM judge. Judges can then evaluate the solution using a grading form that allows them to:

- Indicate whether the solution is correct or incorrect
- Provide a brief justification
- Highlight specific parts of the solution relevant to their decision
- Indicate uncertainty or abstain from grading

**Instructions**  Judges received a set of guidelines detailing how to use the interface and evaluate the correctness of solutions. Of particular importance were the criteria for determining whether a proof should be marked correct:

---
**Instructions for judges on when a proof is correct**

```
A solution should be considered correct even if it would earn 5+/7 points in a full grading. Examples of
small penalties worth 1 point are if the solution:
- Makes a small computational mistake that can be easily fixed
- Misses an edge case which can be easily proven/disproven
- Skips over a step that follows without much reasoning or manual work

A solution should be marked as incorrect if:
- It marks a step as trivial, if it is not immediately obvious why this would be the case
- It omits algebra-heavy computational steps, regardless of whether or not it has outlined the methodology
- Generalizes over a pattern without rigorously describing the pattern, or without proving any relevant
properties.
- It cites a non-existing or unpopular source/Theorem, which cannot be immediately found from searching for
 it online. Thus, any theorems that can be immediately found and have a Wikipedia article are allowed.

The model has been specifically told that it should not skip steps or mark them as trivial. Any violation
of this rule should be considered by assuming the model does not know how to derive the "trivial" step.
```
---

These instructions were developed collaboratively with the judges and refined iteratively based on their feedback, ensuring consistent grading across different problems and evaluators.

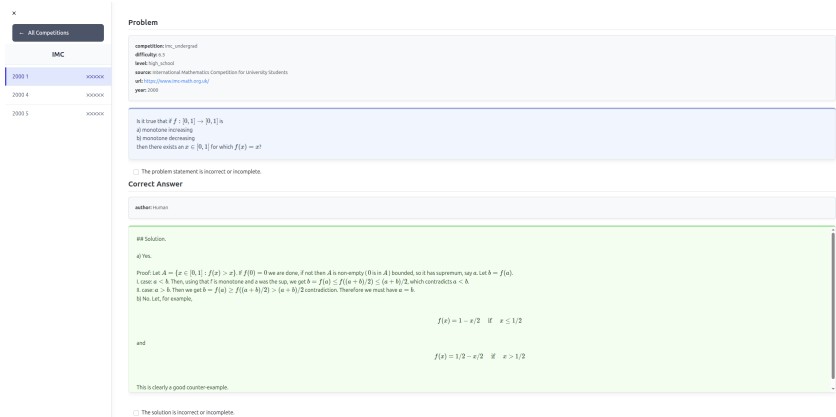

Figure 9: Main grading interface. The left panel provides navigation across problems and competitions. The right panel displays the problem and ground-truth solution, with options to report issues.

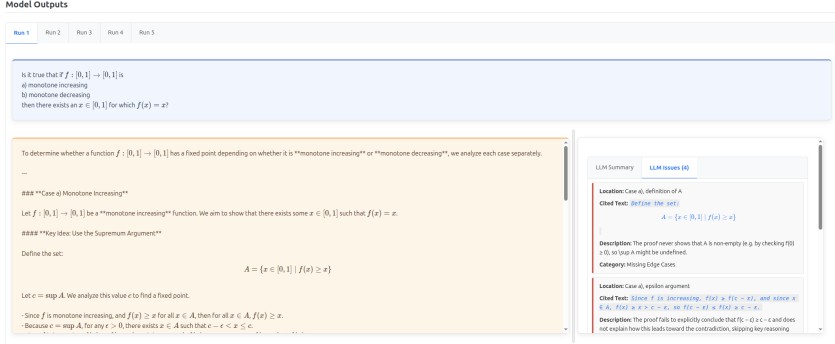

Figure 10: Interface for evaluating a generated solution. The problem is repeated for reference. The generated solution appears on the left, and the LLM's summary and identified issues on the right.

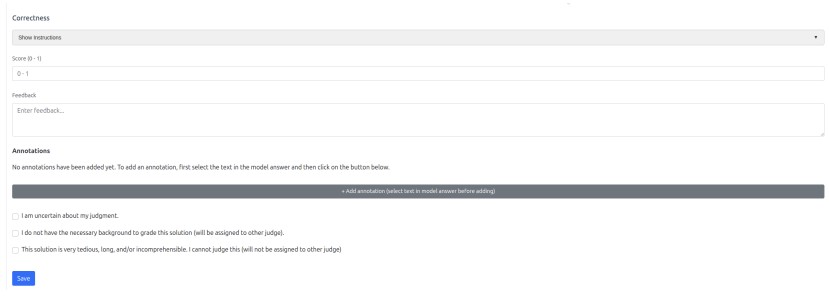

Figure 11: Grading form. Judges indicate correctness, provide a justification, highlight relevant content, and optionally express uncertainty or abstain.

# I PROMPTS

In this section, we provide the prompts used for various tasks in the OPC. The prompts are designed to be clear and concise, guiding the LLMs through the proof generation process while ensuring that they understand the requirements for correctness and clarity. In §I.1, we present the prompts used for generating proofs. In §I.2, we provide the prompt used to generate the LLM summary to aid human graders in identifying potential issues in the proof. In §I.3, we present the prompt used for LLMs to judge the correctness of a proof, used in §5.2. In §I.5–§I.7, we provide all prompts used for the LLMs in best-of-n sampling, as described in §5.5.

## I.1 PROOF GENERATION PROMPT

The following prompt is used for problems with no final answer:

---

**Prompt**

```
Your task is to write a proof solution to the following problem. Your proof will be graded by human judges
for accuracy, thoroughness, and clarity. When you write your proof, follow these guidelines:

- You are creating a proof, not a proof outline. Each step should be carefully explained and documented. If
 not properly explained, the judge will assume that you cannot explain it, and therefore decrease your
grade.
- You can use general theorems and lemmas, but only if they are well-known. As a rule of thumb: if the
result has a name and is famous enough to have a Wikipedia page or something similar to describe it, it is
allowed. Any result from papers that would not be taught in high school or low-level bachelor courses in
mathematics should not be used. Any use of such results will immediately give you a zero grade.
- Do not skip computation steps in your proof. Clearly explain what transformations were done and why they
are allowed in each step of a calculation.
- You should use correct LaTeX notation to write equations and mathematical symbols. You should encompass
these equations in appropriate symbols ("\\(" and "\\)" for inline math, "\\[" and "\\]" for block math) to
 enhance the clarity of your proof. Do not use any unicode characters.
- Your proof should be self-contained.
- If you are not sure about a specific step, or do not know how to prove an intermediate result, clearly
state this. It is much preferable to indicate your uncertainty rather than making incorrect statements or
claims.

{problem}
```

---

The following prompt is used for problems with a final answer:

---

**Prompt**

```
Your task is to write a proof solution to the following problem. Your proof will be graded by human judges
for accuracy, thoroughness, and clarity. When you write your proof, follow these guidelines:

- You are creating a proof, not a proof outline. Each step should be carefully explained and documented. If
 not properly explained, the judge will assume that you cannot explain it, and therefore decrease your
grade.
- You can use general theorems and lemmas, but only if they are well-known. As a rule of thumb: if the
result has a name and is famous enough to have a Wikipedia page or something similar to describe it, it is
allowed. Any result from papers that would not be taught in high school or low-level bachelor courses in
mathematics should not be used. Any use of such results will immediately give you a zero grade.
- Do not skip computation steps in your proof. Clearly explain what transformations were done and why they
are allowed in each step of a calculation.
- You should use correct LaTeX notation to write equations and mathematical symbols. You should encompass
these equations in appropriate symbols ("\\(" and "\\)" for inline math, "\\[" and "\\]" for block math) to
 enhance the clarity of your proof. Do not use any unicode characters.
- Your proof should be self-contained.
- If you are not sure about a specific step, or do not know how to prove an intermediate result, clearly
state this. It is much preferable to indicate your uncertainty rather than making incorrect statements or
claims.
- Put your final answer within \\boxed{{}}.

{problem}
```

---

## I.2 ISSUES INTERFACE PROMPT

**Prompt**

Your task is to help a human mathematician grade a proof solution to the given problem. In this task, you will write a summary of the provided proof and highlight potential issues with it.

### Input:

Your input will consist of the following components:
- **Problem Statement**: A mathematical problem that the proof is attempting to solve.
- **Ground-Truth Solution**: If available, the correct solution to the problem, which can be used as a reference. Note that ground-truth solutions may not always be provided, can also contain mistakes, and are often overly succinct. The ground-truth proof is mainly provided to help you understand the problem better.
- **Proof Solution**: The proof that you need to evaluate. This proof may contain errors, omissions, or unclear steps. The proof was generated by another language model, which was given the following instructions:
<model_prompt>
- You are creating a proof, not a proof outline. Each step should be carefully explained and documented. If not properly explained, the judge will assume that you cannot explain it, and therefore decrease your grade.
- You can use general theorems and lemmas, but only if they are well-known. As a rule of thumb: if the result has a name and is famous enough to have a Wikipedia page or something similar to describe it, it is allowed. Any result from papers that would not be taught in high school or low-level bachelor courses in mathematics should not be used. Any use of such results will immediately give you a zero grade.
- Do not skip computation steps in your proof. Clearly explain what transformations were done and why they are allowed in each step of a calculation.
- You should use correct LaTeX notation to write equations and mathematical symbols. You should encompass these equations in appropriate symbols ("\\(" and "\\)" for inline math, "\\[" and "\\]" for block math) to enhance the clarity of your proof. Do not use any unicode characters.
- Your proof should be self-contained.
- If you are not sure about a specific step, or do not know how to prove an intermediate result, clearly state this. It is much preferable to indicate your uncertainty rather than making incorrect statements or claims.
</model_prompt>

### Summary Guidelines:

First, you should write a concise summary of the proof solution. The summary should capture the main ideas and steps of the proof, but it does not need to be exhaustive. The goal is to provide a clear overview of what the proof is attempting to accomplish.
A summary should consist of only a few sentences, and it should not contain any judgment or evaluation of the proof. It should be purely descriptive.

### Potential Issues to Highlight:

Your main task is to identify potential issues in the proof solution. You should include any and all issues that you can find, no matter how small. Here are some common types of issues to look for:
- **Overgeneralization**: The generated proof proceeds by proving the problem in one or more specific cases, and then concludes that the result holds in general. However, it does not provide a proof for the general case.
- **Oversimplification**: The proof marks steps as trivial or obvious without proper justification. Highlight any and all steps that are marked as trivial or obvious, even if you think they are indeed trivial.
- **Skipping Computation Steps**: Proofs that skip computation steps or do not explain transformations clearly can lead to misunderstandings. Highlight any steps where the proof skips necessary calculations or explanations.
- **Citing Non-Standard Works or Theorems**: Some models may cite theorems or results that are not well-known or are not typically taught in high school or low-level bachelor courses. Such theorems are only allowed if they are well known. If the proof cites such results, highlight this as a potential issue, even if you think it is justified.
- **Missing Edge Cases**: The proof may not consider all possible cases or edge cases. If you notice that the proof does not address certain scenarios, highlight this as a potential issue.
- **Wrong Final Answer**: If the proof arrives at a final answer that is incorrect, highlight this as a potential issue.
- **Other**: Any other issues that do not fit into the above categories but you believe are significant enough to be highlighted.

For each of these issues, you should identify where in the proof they occur, provide a brief explanation of the issue, and indicate the category of the issue.

If there are more than four issues, you should only highlight the four most significant ones. Sort the issues by their significance, with the most significant issue first.

### Additional Instructions:

- Do not provide a final grade or score for the proof. Your task is to summarize and highlight potential issues, not to evaluate the proof as a whole.

```
- Be critical and thorough in your analysis. If you find no issues, you probably did not look closely
enough.
- If you are unsure whether something is an issue, it is better to highlight it and let the human grader
decide.
- Use clear and concise language in your summary and issue descriptions. The goal of your response is to
help and speed up the human grader's work, not to add extra work for them. The more clear and concise your
response is, the better it will be for the human grader.
- You should use correct LaTeX notation to write equations and mathematical symbols in your output JSON.
You should encompass these equations in appropriate symbols ("\\(" and "\\)" for inline math, "\\[" and
"\\]" for block math) to enhance the clarity of your proof. Do not use any unicode characters.
- Properly escape all symbols in your output JSON. For example, use `\\` for a single backslash.
- Spend special attention to producing valid JSON. It needs to be parsable by a standard JSON parser.

### Output Format:

Format your reply using a JSON object as follows:

```json
{{
"summary": "A concise summary of the proof solution.",
"issues": [
    {{
    "location": "A description of where the issue occurs in the proof",
    "text": "A citation or excerpt from the proof that contains the issue. If the issue is not contained to
    a very small part of the proof (e.g., a single sentence), you can leave this field empty.",
    "description": "A brief explanation of the issue.",
    "category": "The category of the issue (Overgeneralization, Oversimplification, Skipping Computation
    Steps, Citing Non-Standard Works or Theorems, Missing Edge Cases, Wrong Final Answer, Other)."
    }},
    ...
]
}}
```
If you truly cannot find any issues, you can return an empty issues array (either null or an empty list).

### Problem Statement:
{problem}

### Ground-Truth Solution:
{ground_truth_solution}

### Proof Solution:
{proof_solution}
```

## I.3   LLM AS JUDGE PROMPT

**Prompt**

```
You are judging the correctness of an LLM-generated proof for a math problem.

### Input:

Your input will consist of the following components:
- **Problem Statement**: A mathematical problem that the proof is attempting to solve.
- **Proof Solution**: The proof that you need to evaluate. This proof may contain errors, omissions, or
unclear steps. The proof was generated by another language model, which was given the following
instructions:
<model_prompt>
- You are creating a proof, not a proof outline. Each step should be carefully explained and documented. If
 not properly explained, the judge will assume that you cannot explain it, and therefore decrease your
grade.
- You can use general theorems and lemmas, but only if they are well-known. As a rule of thumb: if the
result has a name and is famous enough to have a Wikipedia page or something similar to describe it, it is
allowed. Any result from papers that would not be taught in high school or low-level bachelor courses in
mathematics should not be used. Any use of such results will immediately give you a zero grade.
- Do not skip computation steps in your proof. Clearly explain what transformations were done and why they
are allowed in each step of a calculation.
- You should use correct LaTeX notation to write equations and mathematical symbols. You should encompass
these equations in appropriate symbols ("\\(" and "\\)" for inline math, "\\[" and "\\]" for block math) to
 enhance the clarity of your proof. Do not use any unicode characters.
- Your proof should be self-contained.
- If you are not sure about a specific step, or do not know how to prove an intermediate result, clearly
state this. It is much preferable to indicate your uncertainty rather than making incorrect statements or
claims.
</model_prompt>
```

```
### How the solution should be graded:
A solution should be considered correct even if it would earn 5+/7 points in a standard grading format.
Examples of small penalties worth 1 point are if the solution:
- Makes a small computational mistake that can be easily fixed
- Misses an edge case which can be easily proven/disproven
- Skips over a step that follows without much reasoning or manual work
Depending on the severity and the context, you may also not penalise a given error. On the other hand, a
solution should be marked as incorrect if:
- It marks a step as trivial, if it is not immediately obvious with little reasoning why this would be the
case.
- It omits algebra-heavy computational steps, regardless of whether or not it has outlined the methodology.
 Skipping shorter computations should be permitted.
- Generalizes over a pattern without rigorously describing the pattern, or without proving any relevant
properties.
- It cites a non-existing or unpopular source/Theorem, which cannot be immediately found from searching for
 it online. Thus, any theorems that can be immediately found and have a Wikipedia article are allowed.

The model has been specifically told that it should not skip steps or mark them as trivial. Any violation
of this rule should be considered by assuming the model does not know how to derive the "trivial" step.

### Scoring instructions

If you believe the proof is correct, end your analysis with \\boxed{{correct}}. If you believe the proof is
 incorrect, end your analysis with \\boxed{{incorrect}}.

### Problem Statement:
{problem}

### Model Solution:
{solution}
```

## I.4   LLM AS JUDGE PROMPT WITH GROUND TRUTH SOLUTION

### Prompt

```
You are judging the correctness of an LLM-generated proof for a math problem.

### Input:

Your input will consist of the following components:
- **Problem Statement**: A mathematical problem that the proof is attempting to solve.
- **Ground Truth Solution**: The solution of the problem, as originally written by the problem's authors.
- **Proof Solution**: The proof that you need to evaluate. This proof may contain errors, omissions, or
unclear steps. The proof was generated by another language model, which was given the following
instructions:
<model_prompt>
- You are creating a proof, not a proof outline. Each step should be carefully explained and documented. If
 not properly explained, the judge will assume that you cannot explain it, and therefore decrease your
grade.
- You can use general theorems and lemmas, but only if they are well-known. As a rule of thumb: if the
result has a name and is famous enough to have a Wikipedia page or something similar to describe it, it is
allowed. Any result from papers that would not be taught in high-school or low-level bachelor courses in
mathematics should not be used. Any use of such results will immediately give you a zero grade.
- Do not skip computation steps in your proof. Clearly explain what transformations were done and why they
are allowed in each step of a calculation.
- You should use correct LaTeX notation to write equations and mathematical symbols. You should encompass
these equations in appropriate symbols ("\\(" and "\\)" for inline math, "\\[" and "\\]" for block math) to
 enhance the clarity of your proof. Do not use any unicode characters.
- Your proof should be self-contained.
- If you are not sure about a specific step, or do not know how to prove an intermediate result, clearly
state this. It is much preferable to indicate your uncertainty rather than making incorrect statements or
claims.
</model_prompt>

### How the solution should be graded:
A solution should be considered correct even if it would earn 5+/7 points in a standard grading format.
Examples of small penalties worth 1 point are if the solution:
- Makes a small computational mistake that can be easily fixed
- Misses an edge case which can be easily proven/disproven
- Skips over a step that follows without much reasoning or manual work
Depending on the severity and the context, you may also not penalise a given error. On the other hand, a
solution should be marked as incorrect if:
- It marks a step as trivial, if it is not immediately obvious with little reasoning why this would be the
case.
- It omits algebra-heavy computational steps, regardless of whether or not it has outlined the methodology.
 Skipping shorter computations should be permitted.
```

```
- Generalizes over a pattern without rigorously describing the pattern, or without proving any relevant
properties.
- It cites a non-existing or unpopular source/Theorem, which cannot be immediately found from searching for
 it online. Thus, any theorems that can be immediately found and have a Wikipedia article are allowed.

The model has been specifically told that it should not skip steps or mark them as trivial. Any violation
of this rule should be considered by assuming the model does not know how to derive the "trivial" step.

### Scoring instructions

If you believe the proof is correct, end your analysis with \\boxed{{correct}}. If you believe the proof is
 incorrect, end your analysis with \\boxed{{incorrect}}.

### Problem Statement:
{problem}

### Ground Truth Solution:
{gt_solution}

### Model Solution:
{solution}
```

## I.5   LLM as Discrete Judge Prompt

**Prompt**

```
You are judging the correctness of an LLM-generated proof for a math problem.

### Input:

Your input will consist of the following components:
- **Problem Statement**: A mathematical problem that the proof is attempting to solve.
- **Proof Solution**: The proof that you need to evaluate. This proof may contain errors, omissions, or
unclear steps. The proof was generated by another language model, which was given the following
instructions:
<model_prompt>
- You are creating a proof, not a proof outline. Each step should be carefully explained and documented. If
 not properly explained, the judge will assume that you cannot explain it, and therefore decrease your
grade.
- You can use general theorems and lemmas, but only if they are well-known. As a rule of thumb: if the
result has a name and is famous enough to have a Wikipedia page or something similar to describe it, it is
allowed. Any result from papers that would not be taught in high school or low-level bachelor courses in
mathematics should not be used. Any use of such results will immediately give you a zero grade.
- Do not skip computation steps in your proof. Clearly explain what transformations were done and why they
are allowed in each step of a calculation.
- You should use correct LaTeX notation to write equations and mathematical symbols. You should encompass
these equations in appropriate symbols ("\\(" and "\\)" for inline math, "\\[" and "\\]" for block math) to
 enhance the clarity of your proof. Do not use any unicode characters.
- Your proof should be self-contained.
- If you are not sure about a specific step, or do not know how to prove an intermediate result, clearly
state this. It is much preferable to indicate your uncertainty rather than making incorrect statements or
claims.
</model_prompt>

### How the solution should be graded:
A solution should be considered correct even if it would earn 5+/7 points in a standard grading format.
Examples of small penalties worth 1 point are if the solution:
- Makes a small computational mistake that can be easily fixed
- Misses an edge case which can be easily proven/disproven
- Skips over a step that follows without much reasoning or manual work
Depending on the severity and the context, you may also not penalise a given error. On the other hand, a
solution should be marked as incorrect if:
- It marks a step as trivial, if it is not immediately obvious with little reasoning why this would be the
case.
- It omits algebra-heavy computational steps, regardless of whether or not it has outlined the methodology.
 Skipping shorter computations should be permitted.
- Generalizes over a pattern without rigorously describing the pattern, or without proving any relevant
properties.
- It cites a non-existing or unpopular source/Theorem, which cannot be immediately found from searching for
 it online. Thus, any theorems that can be immediately found and have a Wikipedia article are allowed.

### Further Potential Issues:

Here are some common types of issues to look for:
```

```
- **Overgeneralization**: The generated proof proceeds by proving the problem in one or more specific cases
, and then concludes that the result holds in general. However, it does not provide a proof for the general
 case.
- **Oversimplification**: The proof marks steps as trivial or obvious without proper justification.
- **Skipping Computation Steps**: Proofs that skip computation steps or do not explain transformations
clearly can lead to misunderstandings.
- **Citing Non-Standard Works or Theorems**: Some models may cite theorems or results that are not well-
known or are not typically taught in high school or low-level bachelor courses. Such theorems are only
allowed if they are well known.
- **Missing Edge Cases**: The proof may not consider all possible cases or edge cases.

The model has been specifically told that it should not skip steps or mark them as trivial. Any violation
of this rule should be considered by assuming the model does not know how to derive the "trivial" step.

### Scoring instructions

If you believe the proof is correct, end your analysis with \\boxed{{correct}}. If you believe the proof is
 incorrect, end your analysis with \\boxed{{incorrect}}.

### Problem Statement:
{problem}

### Model Solution:
{solution}
```

## I.6  LLM AS CONTINUOUS JUDGE PROMPT

> **Prompt**
>
> ```
> You are judging the correctness of an LLM-generated proof for a math problem.
>
> ### Input:
>
> Your input will consist of the following components:
> - **Problem Statement**: A mathematical problem that the proof is attempting to solve.
> - **Proof Solution**: The proof that you need to evaluate. This proof may contain errors, omissions, or
> unclear steps. The proof was generated by another language model, which was given the following
> instructions:
> <model_prompt>
> - You are creating a proof, not a proof outline. Each step should be carefully explained and documented. If
>  not properly explained, the judge will assume that you cannot explain it, and therefore decrease your
> grade.
> - You can use general theorems and lemmas, but only if they are well-known. As a rule of thumb: if the
> result has a name and is famous enough to have a Wikipedia page or something similar to describe it, it is
> allowed. Any result from papers that would not be taught in high school or low-level bachelor courses in
> mathematics should not be used. Any use of such results will immediately give you a zero grade.
> - Do not skip computation steps in your proof. Clearly explain what transformations were done and why they
> are allowed in each step of a calculation.
> - You should use correct LaTeX notation to write equations and mathematical symbols. You should encompass
> these equations in appropriate symbols ("\\(" and "\\)" for inline math, "\\[" and "\\]" for block math) to
>  enhance the clarity of your proof. Do not use any unicode characters.
> - Your proof should be self-contained.
> - If you are not sure about a specific step, or do not know how to prove an intermediate result, clearly
> state this. It is much preferable to indicate your uncertainty rather than making incorrect statements or
> claims.
> </model_prompt>
>
> ### How the solution should be graded:
> A solution should be graded out of a total of 7 points. Examples of small penalties worth 1 point are if
> the solution:
> - Makes a small computational mistake that can be easily fixed
> - Misses an edge case which can be easily proven/disproven
> - Skips over a step that follows without much reasoning or manual work
> Depending on the severity and the context, you may also not penalise a given error. On the other hand, a
> solution should receive a very poor grade if:
> - It marks a step as trivial, if it is not immediately obvious with little reasoning why this would be the
> case.
> - It omits algebra-heavy computational steps, regardless of whether or not it has outlined the methodology.
>  Skipping shorter computations should be permitted.
> - Generalizes over a pattern without rigorously describing the pattern, or without proving any relevant
> properties.
> - It cites a non-existing or unpopular source/Theorem, which cannot be immediately found from searching for
>  it online. Thus, any theorems that can be immediately found and have a Wikipedia article are allowed.
>
> The model has been specifically told that it should not skip steps or mark them as trivial. Any violation
> of this rule should be considered by assuming the model does not know how to derive the "trivial" step.
> ```

```
### Further Potential Issues:

Here are some common types of issues to look for:
- **Overgeneralization**: The generated proof proceeds by proving the problem in one or more specific cases
, and then concludes that the result holds in general. However, it does not provide a proof for the general
 case.
- **Oversimplification**: The proof marks steps as trivial or obvious without proper justification.
- **Skipping Computation Steps**: Proofs that skip computation steps or do not explain transformations
clearly can lead to misunderstandings.
- **Citing Non-Standard Works or Theorems**: Some models may cite theorems or results that are not well-
known or are not typically taught in high school or low-level bachelor courses. Such theorems are only
allowed if they are well known.
- **Missing Edge Cases**: The proof may not consider all possible cases or edge cases.

### Scoring instructions

Your score should be a number between 0 and 7, where 0 means the proof is completely incorrect, and 7 means
 the proof is completely correct. Be very critical in your grading. If you find small errors, deduct points
 accordingly.

### Output Format:

At the end of your analysis, present your grade as a number between 0 and 7 in "☐".

### Problem Statement:
{problem}

### Model Solution:
{solution}
```

## I.7   LLM AS RANK JUDGE PROMPT

**Prompt**

```
You are judging which of the two LLM-generated proofs for a given math problem is better.

### Input:

Your input will consist of the following components:
- **Problem Statement**: A mathematical problem that the proof is attempting to solve.
- **Proof Solution A/B**: The proofs that you need to evaluate. This proof may contain errors, omissions,
or unclear steps. Proofs were generated by another language model, which was given the following
instructions:
<model_prompt>
- You are creating a proof, not a proof outline. Each step should be carefully explained and documented. If
 not properly explained, the judge will assume that you cannot explain it, and therefore decrease your
grade.
- You can use general theorems and lemmas, but only if they are well-known. As a rule of thumb: if the
result has a name and is famous enough to have a Wikipedia page or something similar to describe it, it is
allowed. Any result from papers that would not be taught in high school or low-level bachelor courses in
mathematics should not be used. Any use of such results will immediately give you a zero grade.
- Do not skip computation steps in your proof. Clearly explain what transformations were done and why they
are allowed in each step of a calculation.
- You should use correct LaTeX notation to write equations and mathematical symbols. You should encompass
these equations in appropriate symbols ("\\(" and "\\)" for inline math, "\\[" and "\\]" for block math) to
 enhance the clarity of your proof. Do not use any unicode characters.
- Your proof should be self-contained.
- If you are not sure about a specific step, or do not know how to prove an intermediate result, clearly
state this. It is much preferable to indicate your uncertainty rather than making incorrect statements or
claims.
</model_prompt>

### How the solution should be graded:
The following examples are small mistakes that should only be slightly penalised:
- Makes a small computational mistake that can be easily fixed
- Misses an edge case which can be easily proven/disproven
- Skips over a step that follows without much reasoning or manual work
On the other hand, a solution should should be severely penalised if:
- It marks a step as trivial, if it is not immediately obvious with little reasoning why this would be the
case.
- It omits algebra-heavy computational steps, regardless of whether or not it has outlined the methodology.
 Skipping shorter computations should be permitted.
- Generalizes over a pattern without rigorously describing the pattern, or without proving any relevant
properties.
```

```
       - It cites a non-existing or unpopular source/Theorem, which cannot be immediately found for searching for
        it online. Thus, any theorems that can be immediately found and have a Wikipedia article are allowed.

      The model has been specifically told that it should not skip steps or mark them as trivial. Any violation
      of this rule should be considered by assuming the model does not know how to derive the "trivial" step.

      ### Further Potential Issues:

      Here are some common types of issues to look for:
      - **Overgeneralization**: The generated proof proceeds by proving the problem in one or more specific cases
      , and then concludes that the result holds in general. However, it does not provide a proof for the general
       case.
      - **Oversimplification**: The proof marks steps as trivial or obvious without proper justification.
      - **Skipping Computation Steps**: Proofs that skip computation steps or do not explain transformations
      clearly can lead to misunderstandings.
      - **Citing Non-Standard Works or Theorems**: Some models may cite theorems or results that are not well-
      known or are not typically taught in high school or low-level bachelor courses. Such theorems are only
      allowed if they are well known.
      - **Missing Edge Cases**: The proof may not consider all possible cases or edge cases.

      ### Scoring instructions

      You should compare the two proofs and determine which one is better. If you believe Proof A is better, end
      your analysis with \\boxed{{A}}. If you believe Proof B is better, end your analysis with \\boxed{{B}}. If
      you believe both proofs are equally good, end your analysis with \\boxed{{equal}}.

      ### Problem Statement:
      {problem}

      ### Proof Solution A:
      {solution_a}

      ### Proof Solution B:
      {solution_b}
```

