# OpenReview forum: "The Open Proof Corpus: A Large-Scale Study of LLM-Generated Mathematical Proofs"
_ICLR.cc/2026/Conference — ICLR 2026 Poster_

### Official Review · Reviewer_Ybod · 2025-10-30

**Soundness:** 3
**Presentation:** 3
**Contribution:** 3
**Rating:** 6
**Confidence:** 4

**Summary:**

LLMs excel at math final-answer tasks (e.g., AIME) but struggle with rigorous proof generation—critical for research/theorem proving. Existing proof benchmarks are small, outdated, or closed, leaving 3 key questions unaddressed: (1) natural vs. formal proof gap, (2) final-answer vs. proof correctness link, (3) best-of-n strategy impact. Several findings are elaborated in the paper:
- Natural language proofs outperform formal ones (GEMINI-2.5-PRO solves 4x more PutnamBench problems than top formal model).
- Final-answer accuracy is not equal to proof correctness (O3 loses ~30% accuracy when proofs are required, vs. 8% for GEMINI-2.5-PRO).
- Best-of-n pairwise ranking boosts accuracy significantly.
- LLMs match humans in proof judging.
- LLMs rarely admit uncertainty and struggle to judge their own work.

**Strengths:**

- Rigorous proof generation/evaluation pipeline.
- First large, open, human-validated LLM-proof dataset (OPC).
- High-Quality Data: Expert judges (IMO background) ensure reliable labels; diverse, competition-sourced problems.
- Actionable Insights: Quantifies critical gaps (natural vs. formal proofs) and validates best-of-n strategies.
- Openness: OPC and code are open-sourced; transparent methodology for reproducibility. I think this will be a good resource in theorem proving area.

**Weaknesses:**

- Narrow Problem Scope: Most problems are high school-level (IMO/USAMO); few undergraduate/research-level tasks.
- Outdated Provers: GROK-4/GPT-5 are only used as judges, not prover, which misses latest LLM proof capabilities.
- Lack of analysis of why formal theorem provers lag behind natural language counterparts. This is an interesting comparison, it would be great if there can be some deeper analysis.

**Questions:**

See weaknesses.

---

> ### Author Response · Authors · 2025-11-19
> **Rebuttal (Ybod)**
>
> We thank the reviewer for their feedback and are happy that they found our pipeline rigorous and our dataset high-quality, with reliable labels and actionable insights. Below, we address each of the reviewer’s questions and remarks in detail.
>
> **Does it significantly affect the dataset and conclusions that more recent provers like GPT-5 and Grok-4 are not included?**
> No, the exclusion of newer models does not significantly affect the validity or usefulness of our results. Of course, the dataset would be better if it also included these more recent provers. However, most of our findings generalize to more recent LLMs, as the underlying patterns of reasoning and common failure modes are similar. Therefore, any judge model trained on our data should generalize well to errors made by these newer systems as well.
>
> In particular, each of our conclusions was created to ensure continued relevance. For instance, the observed performance gap between formal and informal proving will only widen as newer models get introduced. Further, the conclusions related to best-of-n selection are mostly model-agnostic, with no convincing reason why they would not hold for future models as well. While some results, such as the analysis of final-answer discrepancies, are model-dependent, this trend remains, at the moment, consistent: for example, Grok-4 still tends to produce too short “proofs”, while GPT-5 appears less affected by this issue. Finally, although the results shown in Figure 4 are inherently model-specific, this does not meaningfully limit the broader contribution or the general insights derived from our work.
>
> **Why did the authors not include more advanced mathematical problems?**
> See Q1 of our main reply.
>
> **Why do formal theorem provers lag behind their natural language counterparts?**
> See Q2 of our main reply.

---

### Official Review · Reviewer_kWHN · 2025-10-30

**Soundness:** 2
**Presentation:** 3
**Contribution:** 3
**Rating:** 6
**Confidence:** 4

**Summary:**

In this work the authors curated a new dataset called the open proof corpus, which contains 5,062 LLM-generated proofs of 1,010 distinct problems from math contests. These proofs are all incorporated with manual reviews from human experts. They have also done a lot of researches around this dataset, especially on the proof judging capabilities of different LLMs.

**Strengths:**

* This work answers a lot of interesting questions in this area.

* This paper provides a comprehensive study on the proof evaluation capability of LLM, fulfilled an important blank in the research of AI4Math.

**Weaknesses:**

* The evaluation of proof judgement may be heavily dependent on the judger’s prompt or criteria, so neither the judging accuracy nor the alignment with human graders are accurate enough.

* This work did not cover problems from advanced math or research-level math, where proof problems weigh more importance than math competitions. This limits the contribution of OPC.

* The creation of this OPC heavily depends on manual annotation from experts, which limits the scalability of this work.

**Questions:**

* How are the results in Table 2 (Section 5.2) evaluated actually? I did not find any further descriptions of the settings of these experiments around this chapter. If the performance of LLM judges is evaluated by comparing with human labels, then are they directly comparable with human baselines?

---

> ### Author Response · Authors · 2025-11-19
> **Rebuttal (kWHN)**
>
> We thank the reviewer for their feedback and are happy that they found we answered many interesting questions and that we filled in an important gap in the research area. Below, we address each of the reviewer’s questions and remarks in detail.
>
> **How did you attempt to mitigate LLM/human judge inaccuracies due to the specific prompt or criteria used?**
> Both humans and LLMs were provided with detailed, carefully designed instructions to minimize the risk of miscalibration due to ambiguity. Importantly, the prompts given to human and LLM judges were nearly identical, differing only to accommodate output format and related implementation details.
>
> As a result, comparisons between LLMs and humans are meaningful, as both were evaluated under equivalent conditions. Further, because the instructions are very explicit, differences in intra-model performance can be attributed to the models’ understanding and reasoning capabilities rather than to any problems with the prompt itself. Finally, given that LLMs achieve performance comparable to human judges, it is evident that the instructions are well understood: if they were not, the models’ scores would have been significantly lower.
>
> **Can you clarify the results in Table 2?**
> Human performance is measured as the average inter-human agreement on the 10% subset of the OPC that was double-graded, while model performance is measured on the designated test subset. We have now clarified in the revised paper that these subsets differ. However, since the test set was uniformly sampled, this does not affect the validity of the conclusions.
>
> The ground-truth labels in the test set are the annotations produced by a human judge. Consequently, the inter-human agreement of 90.4% reflects the expected accuracy of an additional human annotator relative to that same set. This makes the comparison directly meaningful: both human and LLM performance are evaluated with respect to labels produced by a single human grader.
>
> **Why did the authors not include more advanced mathematical problems?**
> See Q1 of our main reply.
>
> **Can you comment on the scalability of manual annotation from experts?**
> Expert annotation is costly, but necessary. For the conclusions presented in our paper, the current dataset size is already sufficient. It also supports a small-scale fine-tuning experiment, which, as demonstrated, significantly improves performance, reaching levels comparable to leading models. Consequently, scaling up manual annotation further would yield limited returns for the scope of this study.

---

> > ### Comment · Reviewer_kWHN · 2025-11-26
> >
> > Thanks for the detailed responses, which look reasonable to me. Consdering the contribution and impact on the field, I decide to maintain my rating.

---

### Official Review · Reviewer_LpED · 2025-11-02

**Soundness:** 3
**Presentation:** 2
**Contribution:** 4
**Rating:** 8
**Confidence:** 5

**Summary:**

The paper presents large-scale dataset of high-difficulty math problems which includes both correct/incorrect proofs and profound annotations which are represented via human-expert judgements. Based on the dataset, some exciting insights are proposed.

**Strengths:**

1. A fairly large dataset of math problems is proposed, with its core value lying in expert annotations. Furthermore, both correct and incorrect proofs are included. It is quite unusual for modern math benchmarks and datasets to contain such annotations, making this contribution valuable to the community.
2. Some exciting empirical insights are provided. While I do not feel that the exploration of performance differences between natural and formal proof generation is particularly valuable, since the performance degradation in formal setups is largely expected, the examples illustrating the discrepancy between final-answer correctness and proof correctness are important.
3. The performance of the fine-tuned, moderately sized LLM judge is impressive.

**Weaknesses:**

It might be somewhat subjective, but the presentation quality is low, even considering the number of figures included in the manuscript. While it is understandable that the authors tend to include more content rather than placing all figures in the Appendix, the text in each section is highly granular. I think this weakness could be addressed by rethinking the overall structure. The main focus should be on the dataset itself, which is valuable, while the incremental contributions in the form of interesting observations should either be explored more deeply or moved to the Appendix.

**Questions:**

1. Can you clarify the guidelines used by judges for borderline proofs? How are omissions or shortcuts treated when deciding correctness?
2. Could you provide more insight into common errors when models give correct answers but incorrect proofs? Are these mostly algebraic mistakes, logical gaps, or misapplied theorems?
3. Does this discrepancy vary systematically by problem type or difficulty?

---

> ### Author Response · Authors · 2025-11-19
> **Rebuttal (LpED)**
>
> We thank the reviewer for their feedback and are happy that they found our contribution valuable, our empirical insights exciting, and the performance of our judge model impressive. Below, we address each of the reviewer’s questions and remarks in detail.
>
> **Could the authors provide more details about the dataset in the main section?**
> The conclusions we draw from our dataset were a key motivation for creating it in the first place, and we therefore paid significant attention to them in the main text. We view both the dataset and the resulting analyses as equally important contributions of our work. In our updated paper, we have expanded on some information in Section 3 to clarify some details. In particular, we expanded our discussion of the problem selection, added a paragraph about the developer, and increased the detail in the description of the pilot phase. We would be happy to expand this discussion further if the reviewer could specify which particular aspects they feel are missing or would benefit from further clarification.
>
> **Can you clarify the guidelines used by judges for borderline proofs? How are omissions or shortcuts treated when deciding correctness?**
> Judges were given the option to indicate when they were uncertain about the correctness of a proof. We explicitly encouraged them to use this option for borderline cases, enabling us to account for such uncertainty in our analysis. Since only a small portion of the dataset (approximately 3%) fell into this category, including or excluding these cases had a negligible effect on our overall conclusions, and we therefore included them in the main analysis.
> Further, throughout the data collection process, the coordinator tracked grading discrepancies and found that nearly all arose from one grader overlooking a substantial mistake rather than from differences in judgment on borderline proofs.
>
> **Could you provide more insight into common errors when models give correct answers but incorrect proofs?**
> Yes, we performed additional analysis of our results and found that both difficulty and problem type affect these errors. In particular, this behavior is much more frequent for geometric problems and also increases with difficulty. We have added this analysis in Appendix C.2.
>
> Further, there are three main problematic patterns that models display when making these errors. First, models sometimes produce overly short solutions that omit key algebraic or logical steps, making the final answer appear unjustified. Second, in some cases, models cite external “known facts” without specifying the statement, leaving the reasoning incomplete. Finally, we observed instances where models assume an unproven statement that happens to hold for the particular problem instance but is not true in general. While these assumptions can give the right final result, they do not result in valid proofs.

---

### Official Review · Reviewer_J7sP · 2025-11-03

**Soundness:** 3
**Presentation:** 4
**Contribution:** 3
**Rating:** 8
**Confidence:** 4

**Summary:**

This paper introduces the Open Proof Corpus (OPC), a dataset of 5,062 LLM-generated mathematical proofs across 1,010 problems from prestigious competitions (IMO, USAMO, Putnam), each evaluated by human judges. Using the OPC, the authors demonstrate: (1) informal proof generation outperforms formal by 4x on PutnamBench, (2) significant gaps exist between final-answer accuracy and proof correctness (especially for o3, dropping from 87.6% to 59.5%), and (3) ranking-based best-of-n strategies achieve 47% accuracy versus 26% pass@1. They also fine-tune an 8B model that achieves 88.1% accuracy in judging proofs, approaching GPT-5's performance.

**Strengths:**

- Addresses critical need: First large-scale dataset of human-evaluated LLM proofs, filling a major gap since existing benchmarks focus only on final answers (e.g., AIME, HMMT).
- Methodology: Well-designed grading pipeline using 13 former IMO participants, clear guidelines, double-grading (90.4% agreement), and clever use of LLM-generated issue summaries to aid grading efficiency.
- Significant empirical findings: The 4x gap between informal/formal proof generation and the divergence between answer accuracy and proof correctness are important insights. The ranking-based best-of-n showing 21% absolute improvement is practically valuable.
- High-quality dataset design: Thoughtful splits (MathArena, PutnamBench, best-of-n, generic) enable targeted analyses while preventing test set contamination.

**Weaknesses:**

- Binary evaluation loses information: The "5+/7 points counts as correct" threshold is arbitrary and discards nuanced quality differences that partial credit scoring would capture.
- Unfair formal/informal comparison: Comparing specialized formal proof models against general-purpose LLMs isn't apples-to-apples. The brief mention of Seed-Prover's 50% formal accuracy suggests the gap may be overstated.
- Missing statistical analysis: No confidence intervals, significance tests, or error bars despite sufficient sample sizes.
- Insufficient contamination analysis: Section C.2's comparison of "Standard" vs "Non-standard" competitions is suggestive but not conclusive. The performance differences could be explained by difficulty alone.

**Questions:**

- Why binary labels? Could you release the raw judge feedback to enable partial credit analysis? This would be valuable for understanding proof quality gradients.
- Model failure modes: The observation that only 114/1700 incorrect proofs acknowledged uncertainty is striking. Could you analyze whether this varies by problem difficulty or type?
- Formal proof training: Have you considered fine-tuning informal models on formal proof data to better understand the performance gap?
Competition selection rationale: Was there systematic criteria for choosing these specific competitions over others (e.g., Putnam over Mathcounts)?
- Extending beyond competitions: Have you considered including undergraduate textbook problems or research-level lemmas? What would be needed to extend OPC to these domains?

---

> ### Author Response · Authors · 2025-11-19
> **Rebuttal (J7sP)**
>
> We thank the reviewer for their feedback and are happy that they found our work addresses a critical need with a well-designed methodology and that our empirical findings are significant. Below, we address each of the reviewer’s questions and remarks in detail.
>
> **Why did you opt for binary labels rather than grading proofs on a 7-point scale?**
> Binary evaluations are substantially faster to create than grading on a 7-point scale. To ensure consistent calibration across models, a 7-point scale would require a detailed grading rubric, taking significant effort. For example, a proof containing a clear error can immediately be labeled incorrect under binary grading, while a more granular scheme would require reviewing each proof step to determine how many points to deduct.
>
> Our instructions reference the 5+/7 threshold because it is a well-known standard amongst competition graders. This framing clarifies that a proof does not need to be perfect to count as correct. It also helps align the instructions for human judges, all of whom are experienced in mathematics competitions and understand what a typical 5/7 proof looks like.
>
> **Are the specialized formal models unfairly disadvantaged compared to the general-purpose models?**
> No. In fact, our current evaluation gives specialized formal models an advantage. In particular, Goedel-Prover-V2 was evaluated using pass@192, which requires far more computational resources than the single-pass setup used for our models (assuming comparable model sizes).
>
> It is worth noting that these specialized models are fine-tuned from general-purpose models and are optimized for formal proof generation. Thus, these models outperform their general-purpose counterparts. We aim to compare state-of-the-art models in both categories, which clearly highlights the substantial gap between formal and informal approaches. This result is consistent with the inherent difficulty of formal theorem proving.
>
> We did not include comparisons with SeedProver because its computational requirements differ even more drastically. While SeedProver does not specify exact compute usage, it likely involves thousands of model passes **per problem**. A fair apples-to-apples comparison with this agentic framework would therefore require evaluating general-purpose models also within an agentic framework (e.g., https://arxiv.org/abs/2507.15855).
>
> **How can we understand the performance gap between formal and informal models?**
> See Q2 of our main reply.
>
> **Can we finetune informal models on formal proof data to better understand this gap?**
> Formal proof models are already fine-tuned from general-purpose LLMs. For example, the Goedel-Prover-V2 family is fine-tuned from Qwen-3-8B and Qwen-3-32B.
>
> **What does the contamination analysis tell us about the effect of contamination?**
> We agree that the analysis in Figure 7 cannot completely rule out contamination. It represents a best-effort attempt to verify that contamination is *not* the primary driver of our results, i.e., it cannot account for large biases (e.g., >30%). Moreover, since none of our main conclusions depend critically on contamination effects (as discussed in Section 5.6), we are confident that our results remain robust.
>
> **What would be needed to extend the OPC to more advanced mathematics?**
> See Q1 of our main reply.
>
> **Could you analyze whether acknowledging uncertainty varies by problem difficulty or type?**
> We have now performed an analysis that shows that the ratio of acknowledged uncertainty to incorrect answers is roughly constant across difficulty levels. To examine this, we restricted our analysis to the o3 model, as it is the only model that acknowledges uncertainty in a substantial number of cases. We then grouped its performance by competition and, for each competition, obtained both its accuracy and the ratio of acknowledged uncertainty to incorrect answers. Across competitions, the acknowledgement ratio is close to 25%, regardless of accuracy. This estimate is based on a small sample of about 100 uncertainty acknowledgements, so the analysis should be interpreted cautiously.
>
> We were unable to do a similar analysis by problem type due to missing annotations for most problems.
>
> **Can you add a statistical analysis for your results?**
> Yes, we have added confidence intervals for all our results to the revised version of the paper. Our main conclusions remain valid with these results.
>
> **Were there systematic criteria for choosing specific competitions over others?**
> Our inclusion criteria were:
> - The competition is well-known and produces high-quality problems.
> - The difficulty level aligns with our target of roughly a 50% pass rate.
>
> While we considered many competitions, some were excluded simply because the dataset was already sufficiently large. Further, these criteria are at least somewhat subjective, and given that there are plenty of high-quality competitions, there was no need to specify them further.

---

### Official Review · Reviewer_C4Eu · 2025-11-10

**Soundness:** 3
**Presentation:** 3
**Contribution:** 2
**Rating:** 6
**Confidence:** 4

**Summary:**

The authors introduce a set of approximately 1000 contest math problems, drawn from existing competitions, whose LLM proof is humanly rated (binary), to be used both as an eval set and training set. They use the dataset to assess how correct the proof is compared to the (correct) final answer. The dataset, as a training dataset, is validated by fine-tuning an 8B R1-Qwen model, which is claimed to match GPT5.

**Strengths:**

The problem assessment pipeline is rigorous.

The single fine-tuned model on OPC is a welcome addition that supports OPC.

The dataset size is sufficiently large to allow finetuning.

**Weaknesses:**

- misleading statement: my biggest issue is that apparently not the full dataset was shared, as only USAMO and BMOSL seem to appear in the zipped supplementary (and also for these, not the full dataset? seems quite small), contrary to the claim: "We have included our dataset in the supplementary material, along with detailed descriptions of our methodology and experimental setup to ensure full reproducibility."

- ambiguous claim: "The OPC was specifically designed for broad applicability and downstream usage in proof generation research and is the first to include a substantial number of correct, LLM-generated solutions to problems from prestigious mathematics competitions such as the USAMO and IMO."
As it reads, it is unclear if the authors claim to be the first ever to create such a dataset, or the first to create such a dataset in the more narrow domain of contest math problems (only the latter is correct). Since this is from the **appendix**, I would urge the authors to rewrite.

- missed important prior literature: Probably the first paper on "pure" autograding was https://arxiv.org/abs/2406.10268, and there is (rather similar) follow-up work by these authors https://arxiv.org/html/2502.13337v1  (it would be good to include this in the related work section). But much earlier work exists implicitly in ML even if not marked as autograding, e.g. in 2024 https://arxiv.org/abs/2402.11111 a more detailed methodology for "LMs as evaluators" was derived (see also more papers on prior such literature), or **2021** in the well-known paper https://arxiv.org/abs/2110.14168, which used what they called "verifiers" All this points to an existing body of work on proof judging that is missing from this paper.

- Wrong claim: "Data contamination poses only a minor risk for proof judging, since generated proofs cannot be present in the training data."
I am unsure on what information this claim rests - who is to say that the main LLM companies don't exactly do this? They have their LLMs generate outputs on contest-level problems to ensure that their LLMs can potently act as judges, which can be of use for subsequent pipelines that the companies might use, or in case the public wants to use LLMs as judges, and companies are interested in having their LLMs perform well on publicly known problems. This seems entirely plausible to me, so I believe this statement should be retracted.

- in terms of the results, with some exceptions, the paper seems to reinforce known folklore beliefs about how models performance on mathematics.

- Almost no details are given about fine-tuning on R1 Qwen3-8B. In particular, rivalling the performance on GPT-5 is a dubious claim (presumably a heldout subset of OPC was used for this on which R1 Qwen was not train? details are missing)

**Questions:**

N/A

---

> ### Author Response · Authors · 2025-11-19
> **Rebuttal (C4Eu)**
>
> We thank the reviewer for their feedback and are happy that they found our pipeline rigorous, our fine-tuned model a valuable contribution, and appreciated the scale of our dataset. Below, we address each of the reviewer’s questions and remarks.
>
> **Can you provide the dataset in the supplementary material?**
> Yes, the dataset is already included in our supplementary material, specifically in *data/judged_data.json*. We acknowledge that this was not clearly documented in the README and now explicitly indicate where the full dataset can be found. The samples referenced by the reviewer were provided as examples in the repository, but do not contain all the samples. In our full release, we plan to host the dataset on HuggingFace with complete documentation. We have now also added the dataset documentation (README_data.md) to the supplementary material.
>
> As mentioned in L306-307, samples from SMT 2025 were obtained in private correspondence with the competition organizers, who requested that we postpone publishing until their official release. As soon as we receive permission, we will make these samples publicly available. Importantly, these samples concern only a small fraction of the dataset ($\approx 2\%$) and do not affect the reproducibility or validity of our conclusions. We have updated our reproducibility statement to mention this more explicitly.
>
> **Can the authors clarify the claim that the OPC is the first large dataset of LLM-generated solutions from prestigious mathematics competitions?**
> The reviewer correctly notes that we are not the first to construct a large dataset of LLM-generated solutions. For instance, Frieder et al. contains a dataset with mathematical problems involving earlier generation models (up to GPT-4). We have revised the statement as follows: “The OPC [...] is the first large dataset of LLM-generated solutions to problems from prestigious mathematics competitions such as the USAMO and IMO.” We have rewritten the sentence to explicitly restrict the claim to prestigious mathematics competitions (e.g., USAMO/IMO), to avoid any possible misinterpretation as a broader “first-ever” claim.
>
> **Can the authors clarify the claim that contamination only presents a minor risk for judging proofs?**
> We have revised the statement to say it presents a less significant risk rather than only a minor risk. We believe this softened statement is accurate for two main reasons. First, model solutions in the OPC could not have been included in any external training data, as each proof was generated by us and is unique. This implies that mistakes in proofs need to be judged on a case-by-case basis, thereby mitigating the issue of contamination where the judgment can be completely memorized. Second, Table 2 indicates that memorization does not significantly affect our results: even when given the ground-truth solution, model performance does not significantly improve.
>
> **How do the results of this paper differ from folklore beliefs about model performance in this task?**
> We agree that several of our findings align with intuitive expectations. However, quantifying these effects is the only way to rigorously confirm these suspicions, enable systematic model comparison, and supporting deeper analysis. Further, some of our results were *not* previously suspected, including:
> the large discrepancy between the two types of best-of-n selection,
> the finding that judge performance matches human performance, which was further supported later by Luong et al.
> the substantial variation between final-answer and proof-based performance across models, and
> the unexpectedly large, rather than moderate, gap between formal and informal models.
>
> **Can the authors discuss related work for autograding?**
> Yes. We have now incorporated an additional paragraph in our related work section discussing prior work on automatic grading of proofs and LLM as a judge in general. Our paper extends this line of work by constructing a large-scale dataset for fine-tuning an LLM judge, and by demonstrating that state-of-the-art LLMs already achieve near-human performance on proof grading.
>
> **Could you give more information about the finetuning stage of the trained model?**
> We have added a new appendix (Appendix D) describing the fine-tuning procedure in greater detail, including hyperparameters. As mentioned already in lines 339-348, the training and test subsets are completely disjoint. The test set was never used to tune any hyperparameters, ensuring that our final evaluation accurately reflects the model’s true performance. Therefore, although the result is surprising, the model’s performance closely matches that of GPT-5. As expected, however, its performance degrades on undergraduate-level proofs, a task on which it was not trained (see Table 6).

---

### Author Response · Authors · 2025-11-19
**Rebuttal (Main)**

We thank the reviewers for their feedback and positive scores. In particular, we are happy to hear they found our work fills a major gap (J7sP,kWHN), our methodology rigorous (C4Eu, J7sP, Ybod), our findings interesting and significant (J7sP, LpED,kWHN, Ybod), and our finetuned model a welcome addition (C4Eu, LpED). We have identified two common questions among the reviewers, and will address all other points in the review-specific replies.

**Q1. Why did the authors not include more advanced mathematical problems? (kWHN, Ybod, J7sP)**
Extending our pipeline to handle more advanced mathematics would require considerably more time and resources, making it impractical to construct a dataset comparable in scale to the current OPC. Therefore, our focus on high-school and undergraduate problems is a deliberate choice to increase the dataset size. Many of our conclusions depend on having a large number of problems for statistical validity. We expect qualitatively similar patterns for more advanced problems, though quantifying this is an important future direction.

Concretely, creating an advanced-level dataset would lead to a smaller dataset due to two main difficulties. Accurately grading advanced problems would require recruiting mathematics PhD students, who are less available for large-scale annotation efforts. Further, advanced problems need to be carefully designed to ensure correctness: directly extracting theorems from research papers can lead to incomplete or ill-posed problems, as these theorems typically rely on substantial prior context. Even large, well-funded efforts such as FrontierMath have produced only a few hundred problems, opting for final-answer evaluation instead of proof-based grading due to the high cost of manual annotation.

**Q2. Why do formal theorem provers lag behind their natural language counterparts? (Ybod,J7sP)**
Two main factors explain the performance gap. First, formal theorem proving is inherently much harder than natural language proving. Every reasoning step must be rigorously specified, and even minor syntax errors can lead to invalid proofs. For instance, recent 2025 IMO problems formalized in Lean can exceed 4,000 lines of code, whereas their natural language counterparts are far shorter. Second, the quantity of available formal proof data is orders of magnitude smaller than that for informal text, limiting the ability to train large-scale formal systems, as recently argued in the AlphaProof paper [1]. These factors together explain the observed performance disparity.

Providing a deeper analysis of this comparison is hard. Formal and informal proofs are often structurally different, preventing an accurate one-to-one comparison. Our problem set is also too small to provide statistically valid results across problem categories. To the best of our knowledge, there is limited systematic error analysis for formal provers. However, there are some basic observations, such as a difficulty with combinatorics (since these problems are hard to formalize) [2], and research-level problems (since these do not appear in the training data) [3].

[1] https://www.nature.com/articles/s41586-025-09833-y
[2] https://arxiv.org/abs/2505.03171
[3] https://arxiv.org/abs/2511.02872

---

### Meta-Review · Area_Chair_odPp · 2026-01-08

**Summary:**

Across reviewers, there is strong agreement that this submission makes a meaningful contribution to AI4Math by releasing a large, rigorously LLM-generated Open Proof Corpus (OPC), with expert grading and related analyses that the community can reuse for benchmarking and training.

The main shared concerns focus on scope and generality: multiple reviewers questioned whether limiting the dataset largely to high-school and competition problems leaves out more advanced or research-level mathematics, and whether the reliance on expert manual annotation limits scalability. In addition, reviewers asked for deeper and fairer analysis of the gap between formal-vs-informal models. Complementary concerns were raised about evaluation design and robustness, including potential sensitivity of “LLM-as-judge” results to prompts or criteria, binary vs graded scoring, insufficient statistical reporting, and contamination/reproducibility/claims (dataset availability, wording of “first” claims, related work coverage, and fine-tuning details).

In rebuttal, the authors addressed most of these points: they explained the trade-off between adding more advanced mathematics and maintaining dataset scale, and provided protential reasons behind formal provers lagging informal ones. They clarified the comparability of human vs LLM judging protocols, strengthened reporting with confidence intervals/statistical analyses, expanded experimental details, clarified where the full dataset is located in the supplement, and promise to revise claims/related-work wording accordingly.

Overall, the remaining limitations are primarily about breadth (advanced math coverage), depth of formal-proof error analysis, and how much “new discovery” the current empirical findings provide beyond a well-executed, high-value dataset release. The current reviews and discussions support an accept-as-poster recommendation, with a request that the final version incorporate the clarified claims, added experimental details, and the remaining description refinements (optionally in an appendix).

**Reviewer Concerns:**

Most of the key concerns raised by the reviewers were addressed or clarified in the rebuttal. Only a few issues remain, and while the authors provide reasonable explanations and justifications that make them acceptable, these limitations still narrow the paper’s overall scope and impact:

1.	Dataset scope: The corpus is primarily centered on high-school and competition problems, with limited coverage of more advanced or research-level mathematics. The rebuttal explains the trade-off between increasing difficulty and maintaining scale and annotation quality, but the generality of conclusions is still constrained by this scope.

2.	Depth of analysis on the formal vs. informal gap: The authors provide plausible reasons for why formal theorem provers lag behind informal (natural-language) solvers, which helps interpretation. However, a more systematic diagnostic analysis (e.g., tighter controls, clearer attribution of bottlenecks, or deeper error breakdowns) remains limited, so the discussion is best viewed as explanatory rather than definitive.

3.	Scalability of expert-driven grading and construction: The rebuttal reinforces that expert evaluation is a key strength for reliability, but the reliance on expert annotators remains an inherent bottleneck for scaling, refresh cycles, and long-term expansion.

4.	Limited “new discovery” beyond the dataset release: While the rebuttal improves clarity on claims and experimental details, some concern remains that the paper’s primary value is as high-quality infrastructure (dataset + evaluation protocol + baselines), with relatively fewer novel empirical insights compared to the strength of the resource contribution.

**Reviewer Scores:**

Most reviewers would likely keep their original scores unchanged after a full discussion, since the rebuttal primarily clarifies and addresses concerns without materially changing the core contribution (a high-value dataset + baseline analyses) or the remaining scope limitations. The only likely score movement is:

Reviewer C4Eu: 6 -> 7, as the rebuttal directly resolves this reviewer’s main blockers around dataset availability, claim wording, related-work coverage, contamination framing, and missing details.

---

### Decision · Program_Chairs · 2026-01-26

Accept (Poster)